# Learning Individualized Treatment Rules with Many Treatments: A Supervised Clustering Approach Using Adaptive Fusion

**Haixu Ma**
Department of Statistics and Operations Research
University of North Carolina at Chapel Hill
Chapel Hill, NC 27516
`haixuma@live.unc.edu`

**Donglin Zeng**
Department of Biostatistics
University of North Carolina at Chapel Hill
Chapel Hill, NC 27516
`dzeng@email.unc.edu`

**Yufeng Liu**
Department of Statistics and Operations Research
Department of Genetics
Department of Biostatistics
University of North Carolina at Chapel Hill
Chapel Hill, NC 27516
`yfliu@email.unc.edu`

## Abstract

Learning an optimal Individualized Treatment Rule (ITR) is a very important problem in precision medicine. This paper is concerned with the challenge when the number of treatment arms is large, and some groups of treatments in the large treatment space may work similarly for the patients. Motivated by the recent development of supervised clustering, we propose a novel adaptive fusion based method to cluster the treatments with similar treatment effects together and estimate the optimal ITR simultaneously through a single convex optimization. The problem is formulated as balancing *loss+penalty* terms with a tuning parameter, which allows the entire solution path of the treatment clustering process to be clearly visualized hierarchically. For computation, we propose an efficient algorithm based on accelerated proximal gradient and further conduct a novel group-lasso based algorithm for variable selection to boost the performance. Moreover, we demonstrate the theoretical guarantee of recovering the underlying true clustering structure of the treatments for our method. Finally, we demonstrate the superior performance of our method via both simulations and a real data application on cancer treatment, which may assist the decision making process for doctors.

## 1 Introduction

Data-driven individualized decision making problems have received a lot of attentions in precision medicine [1, 2, 3, 4]. Due to possible significant heterogeneity of treatment effects among individuals, it is necessary for decision makers to precisely tailor the treatment decision rules to different subgroups of individuals rather than simply implement the traditional "one size fits all" medical procedures. The main goal of the paper is to recommend the optimal Individualized Treatment Rule (ITR), mapping from the covariate space based on patients' characteristics to the treatment assignment, that optimizes the expected value of a specified reward, known as the value function [5].

36th Conference on Neural Information Processing Systems (NeurIPS 2022).

There are many existing machine learning based approaches for estimating the optimal ITR. Some methods estimate the treatment effect related functions under some prespecified outcome mean models. The estimated ITR is induced by maximizing the estimated treatment effects conditional on patients' covariates. These methods include Q-learning [5], A-learning [6, 7], weighted ordinary least square based method [8] and D-learning [9]. Some other methods circumvent modeling the treatment effect functions by directly estimating the ITR that maximizes the Inverse Probability Weighting (IPW) version of the value function [10, 11, 12, 13, 14]. To overcome the potential misspecification problems for these methods, [15, 16, 17, 18] proposed the doubly robust IPW methods.

**Problem description.** This paper is concerned with the following two questions: *(1) Given a large number of treatment options but limited observations of some specific treatments, how can we effectively estimate the optimal ITR? (2) Some treatments in the large treatment space may work similarly for patients. How can we identify this homogeneous treatment structure and cluster the treatments with similar treatment effects together to reduce the dimension of the treatment space?*

For question (1), the estimated ITR using either the model-based methods or the IPW-based methods can become inaccurate with large variability and numerical instability due to the insufficient number of observations for certain treatments. For question (2), it is common that the drugs are developed based on intervening the same disease symptoms and mechanisms but may be produced by different pharmaceutical companies. [16] and [19] directly combined the treatments with similar treatment effects into certain classes of treatments based on some *prior knowledge*. In general, it is desirable to propose *data-driven* methods to identify the homogeneous treatment structure automatically. However, to our best of knowledge, few existing methods deal with clustering treatments because most literature only consider binary or a moderate number of treatments.

**Related work and our method.** Our goal is to cluster the heterogeneous treatment effects and estimate the ITR under the Q-learning framework [20, 5]. Some methods have been developed to address the problem of identifying subgroups with homogeneous relationships. In supervised learning, with the special focus on exploring the homogeneous relations between the covariates and response, [21], [22] and [23] estimated the group structure for the regression coefficients of covariates with various fusion penalties. To identify subgroups from a heterogeneous population, [24] and [25] assumed the population comes from a mixture of subgroups with their own distributions and utilized the mixture model analysis to classify the observations. However, the mixture model may cause misspecification problems in practice since it needs to specify the underlying distribution for the data and the number of mixture components in the population. Moreover, these supervised learning methods only utilized the information of response and covariates without considering the treatment assignment. Hence, they are not applicable to the ITR problem. In unsupervised learning, clustering analysis is a popular tool. It is usually used to detect the similarities of observations using a predefined distance measure. In particular, a popular visualization method is to draw a dendrogram of the hierarchical clustering using a "bottom up" approach [26].

Regarding to our problem of clustering the treatment effects, we would like to adapt the idea of *latent supervised clustering* [27, 28]. We are interested in clustering the relationship characterized by the regression of "response $\sim$ covariates $\times$ treatments" where "$\times$" refers to the interactions between covariates and treatments. To achieve this goal, we model the treatment effects as the regression problem with treatment-specific coefficient vectors. We implement an adaptive pairwise fusion penalty of the $\ell_1-$distance between the treatment-specific coefficient vectors in order to merge these coefficient vectors with similar values together. This is equivalent to clustering the treatments with similar treatment effects. We formulate the clustering process as a convex minimization problem involving *loss + fusion penalty*, with a tuning parameter balancing these two terms. With this convex formulation, we achieve to maximize the goodness of fit for estimating the heterogeneous treatment effects, while at the same time clustering the treatments without the need of prior knowledge.

**Contribution.** The main contributions of our paper can be summarized as follows. (1) We propose the Supervised Clustering approach using the Adaptive Fusion (SCAF) to identify possible homogeneous group structure within the large treatment space and estimate the ITR in a more effective way. (2) Compared with the two-step method [29], we simultaneously cluster the treatments with similar treatment effects and estimate the optimal ITR. Our method combines the unsupervised and supervised learning together within a single convex optimization problem. (3) We do not need to specify the number of treatment clusters and the entire solution path of the clustering process can

be visualized with a new dendrogram by using a "bottom up" approach. (4) Although [27] and [28] also adopted the supervised clustering technique, they focused on identifying the heterogeneous subpopulations and aimed to cluster the subject-specific regression coefficients. In contrast, SCAF only needs to estimate the treatment-specific coefficients and the dimension of the estimated parameters is dramatically reduced. (5) For computation, we propose an effective algorithm to solve the convex minimization problem based on the accelerated proximal gradient algorithm [30]. Specifically, in order to boost the performance of our fusion algorithm, we propose a novel group-lasso based algorithm to classify the covariates into homogeneous variables and heterogeneous variables respectively. Thus, the fusion penalty only needs to be imposed to the heterogeneous variables that have interaction with treatments. (6) For the theoretical study, we prove the consistency of the regression coefficients and demonstrate that SCAF is able to recover the true underlying clustering structure with probability 1. (7) We conduct both simulation studies and a real data analysis on cancer treatment to illustrate the superior performance of SCAF.

## 2   Methodology

We first introduce the ITR problem under the regression-based framework. Consider the training data $(\boldsymbol{z}_i, a_i, y_i)$ for $i = 1, \ldots, n$ as i.i.d. realizations from the joint distribution of $(Z, A, Y)$, where $\boldsymbol{z}_i \in \mathcal{Z} \subseteq \mathbb{R}^p$ denotes the patient's prognostic variables, $a_i \in \mathcal{A} = \{1, 2, \ldots, M\}$ is the treatment assignment, and $y_i \in \mathbb{R}$ is the observed reward for each patient $i$. Let $\big(Y(a)\big)_{a \in \mathcal{A}} \in \mathbb{R}^M$ be the potential outcome. In addition, define the propensity scores of treatment $p(a|\boldsymbol{z}) := \mathbb{P}(A = a | Z = \boldsymbol{z})$ for $a \in \mathcal{A}$. An ITR $D \in \mathcal{D}$ is a map from the covariate space $\mathcal{Z}$ to the treatment space $\mathcal{A}$. Here, $\mathcal{D} \subseteq \mathcal{A}^{\mathcal{Z}}$ is a prespecified ITR class. The value function of an ITR is defined as $\mathcal{V}(D) = \mathbb{E}[Y\big(D(Z)\big)]$. Assuming that a larger reward is better, our goal is to find the optimal ITR $D^* \in \mathcal{D}$ that maximizes the value function, i.e., $D^* \in \arg\max_{D \in \mathcal{D}} \mathcal{V}(D)$.

We assume the following identifiability assumptions are satisfied [31]: (1) consistency: $Y = \sum_{a \in \mathcal{A}} \mathbb{I}[A = a]Y(a)$; (2) no unmeasured confounders: for each $a \in \mathcal{A}, Y(a) \perp\!\!\!\perp A \mid Z$; (3) positivity: $p(a|\boldsymbol{z}) \geqslant \epsilon > 0$ for any $\boldsymbol{z} \in \mathcal{Z}$. Note that the no unmeasured confounders assumption can be relaxed. If there are possible confounding issues as in the observational studies, our proposed method can be further generalized to deal with such issues by using propensity scores. Based on the three identifiability assumptions, the value function can be written as $\mathcal{V}(D) = \mathbb{E}_Z \big[ \sum_{a \in \mathcal{A}} \mathbb{I}[D(Z) = a] \mathbb{E}[Y|Z, A = a] \big]$. Therefore, the optimal ITR $D^*$ can be derived from $D^*(\boldsymbol{z}) \in \arg\max_{a \in \mathcal{A}} \mathbb{E}[Y|Z = \boldsymbol{z}, A = a]$ for each $\boldsymbol{z} \in \mathcal{Z}$. This motivates us to estimate the conditional treatment effect $\mathbb{E}[Y|Z, A]$. Specifically, we consider the following regression model:

$$Y = M_0(Z) + \sum_{a \in \mathcal{A}} \mathbb{I}[A = a]T(Z; \boldsymbol{\zeta}_a) + \epsilon,$$
$$\text{s.t.} \sum_{a \in \mathcal{A}} T(Z; \boldsymbol{\zeta}_a) = 0; \ \mathbb{E}[\epsilon|Z, A] = 0; \ var[\epsilon|Z, A] < +\infty, \tag{1}$$

where the redundant function $M_0(Z)$ is the main effect of treatments, and $T(Z; \boldsymbol{\zeta}_a)$ is the interaction effect between treatment $a$ and the covariates. We assume that the interaction effect for each treatment $a \in \mathcal{A}$ can be characterized by the treatment-specific parameter $\boldsymbol{\zeta}_a \in \mathbb{R}^p$ that has the same dimension as the covariates $Z$. Here, we consider that the covariates $Z$ contain the intercept term so that the treatment-specific effect is included in $T(Z; \boldsymbol{\zeta}_a)$. Note that a sum-to-zero constraint for the interaction terms is assumed for identifiability of the regression function. Given an estimated parameter $\widehat{\boldsymbol{\zeta}} = (\widehat{\boldsymbol{\zeta}}_1^{\mathsf{T}}, \ldots, \widehat{\boldsymbol{\zeta}}_M^{\mathsf{T}})^{\mathsf{T}}$, the estimated ITR can be equivalently written as $\widehat{D}(\boldsymbol{z}) \in \arg\max_{a \in \mathcal{A}} T(\boldsymbol{z}; \widehat{\boldsymbol{\zeta}}_a)$ for each $\boldsymbol{z} \in \mathcal{Z}$.

With Model (1) in place, we propose our SCAF method to cluster the treatments with similar treatment effects into a combined treatment group. To achieve this, we aim to estimate and fuse the $\boldsymbol{\zeta}_a$'s into several groups. The group structure of the $\boldsymbol{\zeta}_a$'s can equivalently represent the clustering results for the treatments. Note that the main effect function $M_0(Z)$ can be estimated from the weighted parametric or nonparametric regression models [32]. More details about estimating the main effect can be found in Appendix A.1 of supplementary materials. Denote the estimation of $M_0$ as $\widehat{M}_0$, and let $\overline{Y}$ be the residual $Y - \widehat{M}_0(Z)$. In this article, we focus on linear interaction effects, i.e., $T(\boldsymbol{z}, \boldsymbol{\zeta}_a) = \boldsymbol{z}^{\mathsf{T}} \boldsymbol{\zeta}_a$. Let $\boldsymbol{\zeta} = (\boldsymbol{\zeta}_1^{\mathsf{T}}, \ldots, \boldsymbol{\zeta}_M^{\mathsf{T}})^{\mathsf{T}}$. Denote the true value of $\boldsymbol{\zeta}$ to be $\boldsymbol{\zeta}^0 = (\boldsymbol{\zeta}_1^{0\mathsf{T}}, \ldots, \boldsymbol{\zeta}_M^{0\mathsf{T}})^{\mathsf{T}} \in \mathbb{R}^{Mp}$, where $\boldsymbol{\zeta}_a^0 \in \mathbb{R}^p$ is the true parameter for treatment $a = 1, \ldots, M$. In order to estimate and cluster $\boldsymbol{\zeta}_a$'s, we consider the following optimization problem by imposing a

pairwise fusion penalty to each pair of the treatment-specific parameters:

$$\min_{\boldsymbol{\zeta}} \left\{ \mathbb{E}_n \big[ \mathcal{L}(\overline{Y}, \textstyle\sum_{a=1}^{M} \mathbb{I}[A = a] Z^\mathsf{T} \boldsymbol{\zeta}_a) \big] + \textstyle\sum_{1 \leqslant l < t \leqslant M} p_{\lambda_n}(\|\boldsymbol{\zeta}_l - \boldsymbol{\zeta}_t\|_1) \right\}, \tag{2}$$

where $\mathcal{L}(\cdot, \cdot)$ is the prespecified loss function to characterize the goodness of fit, $\|\cdot\|_1$ is the $\ell_1$ norm of a vector, $p_{\lambda_n}$ is the penalty function to encourage fusing $\widehat{\boldsymbol{\zeta}}_a$'s into groups, and $\lambda_n$ is the tuning parameter. Here, $\mathbb{E}_n$ denotes the empirical mean of the training data. Specifically, the optimization problem (2) can be interpreted by maximizing the goodness of fit, while at the same time minimizing the heterogeneity among treatments. When the true treatment effects $\boldsymbol{\zeta}_a^0$'s have homogeneous clustering structure, the estimated parameter $\widehat{\boldsymbol{\zeta}}_a$'s are expected to recover this structure.

In practice, one may assume that only certain elements of $Z \in \mathbb{R}^p$ (denoted as $X \in \mathbb{R}^d$) contribute to the interaction term $T(Z, \boldsymbol{\zeta}_a)$ while others (denoted as $V \in \mathbb{R}^{p-d}$) only show up in the main effects $M_0(Z)$ [28]. Without loss of generality, assume the first $d$ components of $Z$ are the heterogeneous variables $X$ and the remaining $p - d$ variables are the homogeneous variables $V$, i.e., $Z = (X^\mathsf{T}, V^\mathsf{T})^\mathsf{T}$. Let the coefficients for treatment $a$ be $\boldsymbol{\zeta}_a = \big( (\zeta_{a,1}, \ldots, \zeta_{a,d}), (\zeta_{a,d+1} \ldots, \zeta_{a,p}) \big)^\mathsf{T} = (\boldsymbol{\beta}_a^\mathsf{T}, \boldsymbol{\gamma}_a^\mathsf{T})^\mathsf{T}$, where $\zeta_{a,k}$ is the regression coefficient of covariate $Z_k$ ($k = 1, \ldots, p$), and $\boldsymbol{\beta}_a$ and $\boldsymbol{\gamma}_a$ are the coefficients for $X$ and $V$ respectively. By removing the main effect $M_0(Z)$ and consider the residual $\overline{Y}$, the true regression coefficient of $V$ for each treatment should be 0, i.e., $\boldsymbol{\gamma}_a^0 = \mathbf{0}$ for all $a \in \mathcal{A}$. In this case, the penalty term should only be imposed on the coefficients of $X$, i.e., $\boldsymbol{\beta}_a$. If prior knowledge of whether the covariate belongs to $X$ or $V$ is unknown, we propose the group-lasso based algorithm in Section 3 to classify $Z$ into homogeneous variables $V$ and heterogeneous variables $X$. Denote the training data as $(\boldsymbol{x}_i, \boldsymbol{v}_i, a_i, y_i)$ for $i = 1, \ldots, n$, where $\boldsymbol{x}_i \in \mathbb{R}^d$ and $\boldsymbol{v}_i \in \mathbb{R}^{p-d}$. We rewrite Model (1) by decomposing $T(Z; \boldsymbol{\zeta}_a)$ into two parts:

$$\begin{aligned} Y &= M_0(X, V) + \textstyle\sum_{a \in \mathcal{A}} \mathbb{I}[A = a] \big( T(X; \boldsymbol{\beta}_a) + T(V; \boldsymbol{\gamma}_a) \big) + \epsilon \\ &= M_0(X, V) + \textstyle\sum_{a \in \mathcal{A}} \mathbb{I}[A = a] X^\mathsf{T} \boldsymbol{\beta}_a + \epsilon, \\ \text{s.t.} \; &\textstyle\sum_{a \in \mathcal{A}} X^\mathsf{T} \boldsymbol{\beta}_a = 0; \; \mathbb{E}[\epsilon | X, V, A] = 0; \; var[\epsilon | X, V, A] < +\infty. \end{aligned} \tag{3}$$

After distinguishing $X$ and $V$ from the whole variables $Z$, the optimization problem in (2) becomes

$$\widehat{\boldsymbol{\beta}}(\lambda_n) = \operatorname*{arg\,min}_{\boldsymbol{\beta} = (\boldsymbol{\beta}_1^\mathsf{T}, \ldots, \boldsymbol{\beta}_M^\mathsf{T})^\mathsf{T}} \left\{ Q_n(\boldsymbol{\beta}; \lambda_n) := \frac{1}{2} \mathbb{E}_n \big[ \mathcal{L}(\overline{Y}, \sum_{a=1}^{M} \mathbb{I}[A = a] X^\mathsf{T} \boldsymbol{\beta}_a) \big] + \sum_{1 \leqslant l < t \leqslant M} p_{\lambda_n}(\|\boldsymbol{\beta}_l - \boldsymbol{\beta}_t\|_1) \right\}, \tag{4}$$

where $Q_n(\boldsymbol{\beta}; \lambda_n)$ is the optimization function, and $\widehat{\boldsymbol{\beta}}(\lambda_n)$ is the estimated coefficients based on the training data. Note that if we do not detect the homogeneous variables $X$ from $Z$, then the pairwise penalty in (2) would be applied to the $p$-dimensional vector $\boldsymbol{\zeta}_a$'s. For comparison, in (4), only the subvector, i.e., the $d$-dimensional vectors $\boldsymbol{\beta}_a$'s are included in the penalty term. Our empirical results show that our fusing algorithm becomes more effective to solve the optimization problem since the low dimensional vector is easier to be merged together.

The minimization problem (4) can be viewed as the *supervised clustering* process. In the regression-based Q-learning framework, *supervised clustering* can be interpreted by clustering the treatment effects (relationships described by the regression model: reward $Y \sim$ covariates $X \times$ treatments $A$) with the penalized regression problem in (4), which is formulated by balancing *loss + penalty* terms with the tuning parameter $\lambda_n$. The pairwise fusion penalty encourages $\boldsymbol{\beta}_a$'s to merge together so that the treatments with similar treatment effects can be clustered into the same group.

## 3 Algorithms for SCAF

**Group-lasso based algorithm to classify $Z$ into $X$ and $V$.** After obtaining the residual $\overline{Y}$ by subtracting $\widehat{M}_0(Z)$ from $Y$, we propose the following group-lasso based algorithm to identify $X$ and $V$. Recall that $\boldsymbol{\zeta} = (\boldsymbol{\zeta}_1^\mathsf{T}, \ldots, \boldsymbol{\zeta}_M^\mathsf{T})^\mathsf{T}$ and $\boldsymbol{\zeta}_a = (\zeta_{a,1}, \ldots, \zeta_{a,p})^\mathsf{T}$ for treatment $a \in \mathcal{A}$. Note that $\zeta_{a,k}$ is the regression coefficient of covariate $Z_k$ ($k = 1, \ldots, p$) for treatment $a \in \mathcal{A}$. We rewrite $\boldsymbol{\zeta}$ by sorting it based on the covariate order, i.e., $\boldsymbol{\xi} = (\boldsymbol{\xi}_1^\mathsf{T}, \ldots, \boldsymbol{\xi}_p^\mathsf{T})^\mathsf{T}$, where $\boldsymbol{\xi}_k = (\zeta_{1,k}, \ldots, \zeta_{M,k})^\mathsf{T}$ for $k = 1, \ldots, p$, is the coefficient vector of $Z_k$ for all treatments. By the definition of the homogeneous variables $V \in \mathbb{R}^{p-d}$, the true regression coefficient $\zeta_{a,k}^0 = 0$ for all $k = d + 1, \ldots, p$ and all $a \in \mathcal{A}$. Hence, the whole true coefficient vector can be written by the following sparse group structure,

i.e, $\boldsymbol{\xi}^0 = (\boldsymbol{\xi}_1^{0\mathsf{T}}, \ldots, \boldsymbol{\xi}_d^{0\mathsf{T}}, \mathbf{0}^\mathsf{T}, \ldots, \mathbf{0}^\mathsf{T})^\mathsf{T}$, where the coefficients of each covariate are considered as a group. This motivates us to implement group-lasso penalty to shrink the whole group vector $\boldsymbol{\xi}_k$ that corresponds to covariate $Z_k$ to $\mathbf{0}$ to achieve group sparsity. If $\widehat{\boldsymbol{\xi}}_k = \mathbf{0}$, then $Z_k$ should be classified as $V$, otherwise it should belong to $X$. In particular, we consider the following optimization problem:

$$\widehat{\boldsymbol{\xi}} = \arg\min_{\boldsymbol{\xi}} \left\{ \mathbb{E}_n \big[ \textstyle\sum_{a=1}^M \mathbb{I}[A = a](\overline{Y} - \sum_{k=1}^p \zeta_{a,k} Z_k)^2 \big] + \lambda_{\text{glasso}} \textstyle\sum_{k=1}^p \|\boldsymbol{\xi}_k\|_b \right\}, \qquad (5)$$

where $\|\cdot\|_b$ is a given base norm for each $\boldsymbol{\xi}_k$ and $\lambda_{\text{glasso}}$ is the tuning parameter. Typical choices of base norms include the $\|\cdot\|_2$ or $\|\cdot\|_\infty$ norm, so that the group-lasso penalty term becomes the block $\ell_1/\ell_2$ or $\ell_1/\ell_\infty$ norm, i.e., the $\ell_1$ norm on all groups with the $\ell_2$ or $\ell_\infty$ norm for each group.

**Adaptive proximal gradient algorithm.** Next we introduce our main algorithm to solve problem (4). We consider the $\ell_2$-loss function for $\mathcal{L}(\cdot, \cdot)$ and choose the adaptive pairwise fusion penalty weighted by $\boldsymbol{\omega}$ for $p_{\lambda_n}(\cdot)$. Let $\boldsymbol{\beta} = (\boldsymbol{\beta}_1^\mathsf{T}, \ldots, \boldsymbol{\beta}_M^\mathsf{T})^\mathsf{T}$. Specifically, the optimization problem (4) can be written in the following compact form

$$\widehat{\boldsymbol{\beta}}(\lambda_n) = \arg\min_{\boldsymbol{\beta}} \left\{ Q_n(\boldsymbol{\beta}; \lambda_n) := L_n(\boldsymbol{\beta}) + P_n(\boldsymbol{\beta}) \right\}, \qquad (6)$$

where

$$L_n(\boldsymbol{\beta}) := \tfrac{1}{2} \mathbb{E}_n \big[ (\overline{Y} - \textstyle\sum_{a=1}^M \mathbb{I}[A = a] X^\mathsf{T} \boldsymbol{\beta}_a)^2 \big], \ P_n(\boldsymbol{\beta}) := \lambda_n \sum_{1 \leqslant l < t \leqslant M} \boldsymbol{\omega}_{l,t} \|\boldsymbol{\beta}_l - \boldsymbol{\beta}_t\|_1 . \quad (7)$$

Here, $\boldsymbol{\omega}_{l,t}$ is the weight of penalizing the $\ell_1$-distance between $\boldsymbol{\beta}_l$ and $\boldsymbol{\beta}_t$. We iteratively solve the problem coming from (6) and (7). In practice, we find that $\boldsymbol{\omega}_{l,t} = \min\left\{ B_{\boldsymbol{\omega}}, 1/\|\widetilde{\boldsymbol{\beta}}_l - \widetilde{\boldsymbol{\beta}}_t\|_1 \right\}$ can be a good option where $\widetilde{\boldsymbol{\beta}} = (\widetilde{\boldsymbol{\beta}}_1^\mathsf{T}, \ldots, \widetilde{\boldsymbol{\beta}}_M^\mathsf{T})^\mathsf{T}$ is an estimate for $\boldsymbol{\beta}$ from the last iteration. The initial estimate for $\boldsymbol{\beta}$ can be obtained from (5) in the group-lasso step. Moreover, $B_{\boldsymbol{\omega}}$ is a prespecified constant to upper bound $\boldsymbol{\omega}_{l,t}$ to handle small distance of the pair $(\widetilde{\boldsymbol{\beta}}_l, \widetilde{\boldsymbol{\beta}}_t)$. The adaptive weight helps to adjust for the potential bias created by the $\ell_1$-penalty. Intuitively, large weight is expected to be assigned to treatments within the same group, otherwise, small weights are implemented.

Note that both $L_n(\boldsymbol{\beta})$ and $P_n(\boldsymbol{\beta})$ are convex functions for $\boldsymbol{\beta}$, and $L_n(\boldsymbol{\beta})$ has Lipschitz gradient. Thus, we can utilize the accelerated proximal gradient algorithm [30] to iteratively solve the problem coming from (6) and (7). Denote the gradient vector of $L_n(\boldsymbol{\beta})$ as $\nabla L_n(\boldsymbol{\beta})$ and the Lipschitz constant of $\nabla L_n(\boldsymbol{\beta})$ as $l_n$. The computational cost of our algorithm mainly includes two following parts: (1) calculate $l_n$ and evaluate the gradient $\nabla L_n(\boldsymbol{\beta})$ for the updated $\boldsymbol{\beta}$; (2) compute the proximal operator of $P_n$, which is defined as $\text{prox}_{s_n P_n}(\boldsymbol{\beta}) := \arg\min_{\overline{\boldsymbol{\beta}}} \left\{ P_n(\overline{\boldsymbol{\beta}}) + \frac{1}{2s_n} \|\overline{\boldsymbol{\beta}} - \boldsymbol{\beta}\|_2^2 \right\}$, for any updated $\boldsymbol{\beta}$ and the step size $s_n > 0$. For a fixed $\lambda_n$, the main steps of the proposed algorithm for SCAF are summarized in Algorithm 1. The tuning parameter $\lambda_n$ can be tuned by cross validation based on the IPWE of the value function performed on the validation data. For $\boldsymbol{z} \in \mathcal{Z}$, the final estimated ITR $\widehat{D}(\boldsymbol{z})$ is obtained from randomly sampling one treatment from the optimal estimated treatment group with probability proportional to the propensity score. The tuning criteria is $\mathbb{E}_n \big[ \frac{\mathbb{I}(\widehat{D}(Z)=A)}{p(A|Z)} Y \big] / \mathbb{E}_n \big[ \frac{\mathbb{I}(\widehat{D}(Z)=A)}{p(A|Z)} \big]$, which is larger the better [33, 10]. Here, $\mathbb{E}_n$ denotes the empirical mean for the validation data.

## 4 Theoretical properties

In this section, we establish the theoretical guarantee for consistency of the estimated regression coefficients. For Model (3), consider the $M$ treatments can be partitioned into $K$ disjoint treatment groups $\{\mathcal{G}_k\}_{k=1}^K$, where the effects of treatments within the same treatment group are identical. Here, the group number $K$ is typically unknown in practice. Let $\boldsymbol{\beta}^0 = (\boldsymbol{\beta}_1^{0\mathsf{T}}, \boldsymbol{\beta}_2^{0\mathsf{T}}, \ldots, \boldsymbol{\beta}_M^{0\mathsf{T}})^\mathsf{T}$ be the true treatment-specific regression coefficients in (3). The values of $\boldsymbol{\beta}_a^0$'s from the same treatment group $\mathcal{G}_k$ for $k = 1, \ldots, K$ are the same. In particular, we aim to develop the convergence of $\widehat{\boldsymbol{\beta}}(\lambda_n) \to \boldsymbol{\beta}^0$ as $n \to \infty$, which equivalently demonstrates consistency of recovering the true group structure.

Let $\mathbf{X} \in \mathbb{R}^{n \times p}$ be the design matrix of $X$. Denote $\mathbf{X}_a \in \mathbb{R}^{n_a \times p}$ as the submatrix of $\mathbf{X}$ where the observations in $\mathbf{X}_a$ are assigned to treatment $a$. Here, $n_a$ is the number of patients receiving treatment $a$ and hence $\sum_{a=1}^M n_a = n$. Further denote $\mathbf{U} = \text{diag}(\mathbf{X}_1, \mathbf{X}_2, \ldots, \mathbf{X}_M) \in \mathbb{R}^{n \times Mp}$. Under the true group structure, let $\boldsymbol{\alpha}^0 = (\boldsymbol{\alpha}_1^{0\mathsf{T}}, \boldsymbol{\alpha}_2^{0\mathsf{T}}, \ldots, \boldsymbol{\alpha}_K^{0\mathsf{T}})^\mathsf{T} \in \mathbb{R}^{Kp}$ be the distinct values of $\boldsymbol{\beta}^0$, where $\boldsymbol{\alpha}_k^0 \in \mathbb{R}^p$

---

**Algorithm 1: SCAF**

---

**Step 1**: Sort the observations based on the assigned treatment order.
**Step 2**: Remove the main effect $M_0(Z)$ and calculate residual $\bar{y}$.
**Step 3**: Implement group-lasso to identify heterogeneous variables $X$ from $Z$.
**Step 4**: Use adaptive fast proximal gradient algorithm to solve problem (6):
  (1) Obtain the initial point $\boldsymbol{\beta}^{(0)}$ from Step 2 and set the desired tolerance $\epsilon_0 > 0$;
  (2) Compute the Lipschitz constant $l_n = \lambda_{\max}(\mathbf{U}^\intercal\mathbf{U})$ and set the step-size $s_n = 1/l_n$, $t_0 = 1$;
  (3) Let $\widehat{\boldsymbol{\beta}}^{(0)} := \boldsymbol{\beta}^{(0)}$ and set $\boldsymbol{\omega}_{l,t}^{(0)} := \min\{B_{\boldsymbol{\omega}}, 1/\|\widehat{\boldsymbol{\beta}}_l^{(0)} - \widehat{\boldsymbol{\beta}}_t^{(0)}\|_1\}$ for $P_n^{(0)}(\boldsymbol{\beta})$ $(l, t \in \mathcal{A})$;
  (4) For $i = 0, 1, \ldots, i_{\max}$, do:
    a. Compute $\boldsymbol{\beta}^{(i+1)} \approx \text{prox}_{s_n P_n^{(i)}}(\widehat{\boldsymbol{\beta}}^{(i)} - s_n \nabla L_n(\widehat{\boldsymbol{\beta}}^{(i)}))$ [28];
    b. Update $t_{i+1} := (1 + \sqrt{1 + 4t_i^2})/2$;
    c. Perform FISTA [34] with $\widehat{\boldsymbol{\beta}}^{(i+1)} := \boldsymbol{\beta}^{(i+1)} + \frac{t_i - 1}{t_{i+1}}(\boldsymbol{\beta}^{(i+1)} - \boldsymbol{\beta}^{(i)})$;
    d. If $\|\widehat{\boldsymbol{\beta}}^{(i+1)} - \widehat{\boldsymbol{\beta}}^{(i)}\| \leqslant \epsilon_0$, then end the loop;
    e. Update $\boldsymbol{\omega}_{l,t}^{(i+1)} := \min\{B_{\boldsymbol{\omega}}, 1/\|\widehat{\boldsymbol{\beta}}_l^{(i+1)} - \widehat{\boldsymbol{\beta}}_t^{(i+1)}\|_1\}$ for $P_n^{(i+1)}(\boldsymbol{\beta})$ $(l, t \in \mathcal{A})$;
  (5) End of the main loop.
**Step 5**: Obtain the estimated ITR $\widehat{D}(\boldsymbol{x}) \in \arg\max_{a \in \mathcal{A}} \boldsymbol{x}^\intercal \widehat{\boldsymbol{\beta}}_a$ for $\boldsymbol{x} \in \mathcal{X}$.

---

is the true treatment group-specific coefficients for $k = 1, \ldots, K$. Denote $N_k$ to be the number of patients whose assigned treatments belong to treatment group $\mathcal{G}_k$ for $k = 1, \ldots, K$. Then we have $\sum_{k=1}^{K} N_k = n$. In addition, we use $|\mathcal{G}_k|$ to denote the number of treatments in the $k$-th group where $\sum_{k=1}^{K} |\mathcal{G}_k| = M$. For simplicity of notations, we assume that the treatment index has been well sorted based on the true group structure. Then, under the true group memberships $\mathcal{G}_1, \ldots, \mathcal{G}_K$, let $\mathbf{H}_k \in \mathbb{R}^{N_k \times p}$ be the submatrix of $\mathbf{X}$ where the observations in $\mathbf{H}_k$ receive treatments in group $\mathcal{G}_k$ for $k = 1, \ldots, K$. Further denote $\mathbf{H} = \text{diag}(\mathbf{H}_1, \mathbf{H}_2, \ldots, \mathbf{H}_K) \in \mathbb{R}^{n \times Kp}$. In particular, under the true group structure, define the oracle estimator for $\boldsymbol{\alpha}^0$ to be $\widehat{\boldsymbol{\alpha}}^{or} := \arg\min_{\boldsymbol{\alpha}} \frac{1}{2} \|\bar{y} - \mathbf{H}\boldsymbol{\alpha}\|_2^2$. Equivalently, $\widehat{\boldsymbol{\alpha}}^{or} = (\mathbf{H}^\intercal\mathbf{H})^{-1}\mathbf{H}^\intercal\bar{y}$. The corresponding oracle estimator for $\boldsymbol{\beta}$, denoted as $\widehat{\boldsymbol{\beta}}^{or}$, is obtained by expanding $\widehat{\boldsymbol{\alpha}}^{or}$ based on the true group structure.

Hereafter, we allow the number of treatments $M$, the true treatment group number $K$, and the covariates' dimension $p$ to grow as the sample size $n$ increases. We use $M_n$, $K_n$, and $p_n$ to denote them respectively. In addition, for any given matrix $\mathbf{G} = (G_{ij})_{i=1,j=1}^{s,t}$, denote $\|\mathbf{G}\|_\infty := \max_{1 \leqslant i \leqslant s} \sum_{j=1}^{t} |G_{ij}|$. For a vector $g = (g_1, \ldots, g_s)^\intercal \in \mathbb{R}^s$, denote $\|g\|_\infty := \max_{1 \leqslant i \leqslant s} |g_i|$. We use $l_n \gg k_n$ to denote $l_n^{-1} k_n = o(1)$.

We first establish the concentration bound for consistency of the oracle estimator when the observation numbers in each treatment group employ a suitable distribution pattern. Specifically, let $N_{\min} := \min_{k=1,\ldots,K} N_k$ be the minimum number of observations receiving the treatments from the same treatment group. We make the following regularity assumptions in the high-dimensional statistical literature [27, 35]:

**Assumption 1.** Assume $\|\boldsymbol{X}_j\|_2 = \sqrt{n}$ for $j = 1, \ldots, p_n$, and $\|\mathbf{U}^\intercal\mathbf{U}\|_\infty \leqslant B_1 M_n p_n$ where $B_1 > 0$.

**Assumption 2.** The minimum eigen value $\lambda_{\min}(\mathbf{H}^\intercal\mathbf{H}) \geqslant C_1 N_{\min}$ where $C_1 > 0$.

**Assumption 3.** Let $\boldsymbol{\epsilon} = (\epsilon_1, \ldots, \epsilon_n)^\intercal \in \mathbb{R}^n$ be the error vector in model (3). Suppose $\boldsymbol{\epsilon}$ has sub-Gaussian tail bound, i.e., there exists a constant $c_1 \in (0, +\infty)$ such that, for any vector $\boldsymbol{a} \in \mathbb{R}^n$ and any $t > 0$, we have $Pr(|\boldsymbol{a}^\intercal\boldsymbol{\epsilon}| > \|\boldsymbol{a}\|_2 t) < 2\exp(-c_1 t^2)$.

**Theorem 1** (Consistency of oracle estimator). *Suppose Assumptions 1-3 hold. If $K_n p_n = o(n)$, and $N_{min} \gg \sqrt{(K_n p_n)n \log n}$, then with probability at least $1 - 2K_n p_n/n$, we have $\left\|\widehat{\boldsymbol{\beta}}^{or} - \boldsymbol{\beta}^0\right\|_\infty = \left\|\widehat{\boldsymbol{\alpha}}^{or} - \boldsymbol{\alpha}^0\right\|_\infty \leqslant \phi_n$, where $\phi_n = c_1^{-1/2} C_1^{-1} \sqrt{K_n p_n} N_{min}^{-1} \sqrt{n \log n}$.*

Since $\sqrt{(K_n p_n)n \log n} \ll N_{\min} \leqslant n/K_n$, we have $K_n\sqrt{(K_n p_n)} = o(\sqrt{n/\log n})$. Hence, $K_n$ must satisfy $K_n = o(n^{1/3}/(\log n)^{1/3})$. It indicates that we cannot have too many treatment groups. If we set $N_{\min} = \delta n/K_n$ where $\delta \in (0, 1)$, then the bound $\phi_n = c_1^{-1/2} C_1^{-1} \delta^{-1} K_n \sqrt{K_n p_n} \sqrt{\log n/n}$.

Our next step is to show that the oracle estimator $\widehat{\boldsymbol{\beta}}^{or}$ is a local minimizer of the object function $Q_n(\boldsymbol{\beta}; \lambda_n)$ in (6) with probability 1 by giving a lower bound of the minimum signal difference between treatment groups. Then the convergence of $\widehat{\boldsymbol{\beta}}(\lambda_n) \to \boldsymbol{\beta}^0$ can be established. To achieve that, we need to assume the penalty function $p_{\lambda_n}(\cdot)$ in (4) has sharp derivative around 0 to adjust the bias of the fusion penalty. Some popular penalties considered in [36], [37] and [38] satisfy similar properties as the following assumption.

**Assumption 4.** Suppose $p_{\lambda_n}(\cdot)$ is symmetric around 0 for each $\lambda_n$, and there exists a constant $a > 0$ such that $p_{\lambda_n}(t)$ is a constant for all $t \geqslant \frac{a}{2}\lambda_n$. Assume $p_{\lambda_n}(0) = 0$, and it is differentiable around 0 with its derivative satisfying $p'_{\lambda_n}(t) \geqslant \mathcal{O}(\sqrt{n \log n}) / \inf_{1 \leqslant k \leqslant K_n} |\mathcal{G}_k|$ for all $0 < t \leqslant 2\phi_n$.

Let $b_n = \inf_{i \in \mathcal{G}_k, j \in \mathcal{G}_{k'}, k \neq k'} \|\boldsymbol{\beta}_i^0 - \boldsymbol{\beta}_j^0\|_1 = \inf_{k \neq k'} \|\boldsymbol{\alpha}_k^0 - \boldsymbol{\alpha}_{k'}^0\|_1$ be the minimal difference of the regression coefficients between two groups. Intuitively, we need the distances between two different groups to be large enough so that we can recover the group membership. In particular, we make the following assumption. A similar assumption was also used in [27] and [39].

**Assumption 5.** Suppose $b_n / p_n > a\lambda_n$ and $\lambda_n \gg \phi_n$.

**Theorem 2.** *Suppose Assumptions 1-5 hold, then there exists a local minimizer $\widehat{\boldsymbol{\beta}}(\lambda_n)$ of the objective function $Q_n(\boldsymbol{\beta}; \lambda_n)$ in (6) such that $Pr\big(\widehat{\boldsymbol{\beta}}(\lambda_n) = \widehat{\boldsymbol{\beta}}^{or}\big) \to 1$.*

Theorem 1 together with Theorem 2 establish the convergence property of $\widehat{\boldsymbol{\beta}}(\lambda_n) \to \boldsymbol{\beta}^0$. For our implementation in Section 3, we iteratively solve the minimization problem (4) with the adaptive fusion penalty shown in (7). We use $\widehat{\boldsymbol{\beta}}^{(i)} = (\widehat{\boldsymbol{\beta}}_1^{(i)}, \ldots, \widehat{\boldsymbol{\beta}}_M^{(i)})^\intercal$ to denote the estimation of $\boldsymbol{\beta}$ in the $i$-th iteration. For $l, t \in \mathcal{A}$, the weight in the $i$-th iteration $\boldsymbol{\omega}_{l,t}^{(i)}$ is specified as $\min\{B_{\boldsymbol{\omega}}, 1/\|\widehat{\boldsymbol{\beta}}_l^{(i)} - \widehat{\boldsymbol{\beta}}_t^{(i+1)}\|_1\}$. Hence, the derivative of $p_{\lambda_n}(\cdot)$ evaluated at $\|\widehat{\boldsymbol{\beta}}_l^{(i)} - \widehat{\boldsymbol{\beta}}_t^{(i)}\|_1$, which equals to $\lambda_n \boldsymbol{\omega}_{l,t}^{(i)}$, will become extremely large when $\|\widehat{\boldsymbol{\beta}}_l^{(i)} - \widehat{\boldsymbol{\beta}}_t^{(i)}\|_1$ is close to 0, and it will be relatively small if the pair $(\widehat{\boldsymbol{\beta}}_l^{(i)}, \widehat{\boldsymbol{\beta}}_t^{(i)})$ are not fused together. Consequently, if $\widehat{\boldsymbol{\beta}}$ is close to $\boldsymbol{\beta}^0$, then the derivative of $p_{\lambda_n}(\cdot)$ will approximately satisfy the properties in Assumption 4 within a neighbourhood around $\boldsymbol{\beta}^0$. Therefore, based on Theorem 2, the oracle estimator $\widehat{\boldsymbol{\beta}}^{or}$ can still be the local minimizer of $Q_n(\boldsymbol{\beta}; \lambda_n)$ within a neighbourhood around $\boldsymbol{\beta}^0$ when our adaptive fusion penalty is utilized.

## 5 Experiments

We evaluate the finite-sample performance of our method using simulation and a real data application to the Patient-Derived Xenograft (PDX) study. We compare our method with the state-of-art methods for the ITR problem: (a) RF (Random Forest, [40]); (b) PLS ($\ell_1$-Penalized Least Square method, [5]); (c) AD (Multi-armed Angle-based Direct learning method, [9]). However, none of these existing methods consider the treatment group structure in a large treatment space.

**Data generation for simulation.** We generate 10-dimensional independent feature variables $Z_1, \ldots, Z_{10}$, following $U[-1, 1]$. The outcome $Y$ is normally distributed with $\mathbb{E}[Y|Z, A] = 1 + 2Z_1 + Z_2 + 0.5Z_3 + T_0(Z, A)$ and standard deviation 1, where $T_0(Z, A)$ reflects the interaction between the treatment and the feature variables. Specifically, we conduct the following three scenarios where the treatment effects have the homogeneous grouping structure $\mathcal{G}$:

**Scenario 1.** $M = 10, K = 2, \mathcal{G} = \{\mathcal{G}_1, \mathcal{G}_2\} = \{\{1, 2, 3, 4, 5\}, \{6, 7, 8, 9, 10\}\}$, $X = (1, Z_1, Z_2)$, and $T_0(Z, A) = 1.8\big(0.2 - Z_1 - Z_2\big)\big(\mathbb{I}[A \in \mathcal{G}_1] \times (-1) + \mathbb{I}[A \in \mathcal{G}_2] \times 1\big)$;

**Scenario 2.** $M = 15, K = 3, \mathcal{G} = \{\mathcal{G}_1, \mathcal{G}_2, \mathcal{G}_3\} = \{\{1, 2, 3, 4, 5\}, \{6, 7, 8, 9, 10\}, \{11, 12, 13, 14, 15\}\}$, $X = (1, Z_1, Z_2)$, and $T_0(Z, A) = 5\big((-0.2 + Z_1 + 2Z_2)\mathbb{I}[A \in \mathcal{G}_1] + (0.3 + 2Z_1 + Z_2)\mathbb{I}[A \in \mathcal{G}_2] + (-0.2 + 3Z_1)\mathbb{I}[A \in \mathcal{G}_3]\big)$;

**Scenario 3.** $M = 20, K = 5, \mathcal{G} = \{\mathcal{G}_1, \mathcal{G}_2, \mathcal{G}_3, \mathcal{G}_4, \mathcal{G}_5\} = \{\{1, 2, 3, 4\}, \{5, 6, 7, 8\}, \{9, 10, 11, 12\}, \{13, 14, 15, 16\}, \{17, 18, 19, 20\}\}$, $X = (1, Z_1, Z_2, Z_3, Z_4)$, and $T_0(Z, A) = 3.5\big((0.5 + 0.5Z_1 + 1.5Z_2 + 2Z_3 + 1.5Z_4)\mathbb{I}[A \in \mathcal{G}_1] + (1 - 2Z_1 - Z_2 - 3Z_3 - 2Z_4)\mathbb{I}[A \in \mathcal{G}_2] + (-1 + Z_1 - 2Z_2 + Z_3 - Z_4)\mathbb{I}[A \in \mathcal{G}_3] + (-2 - Z_1 + Z_2 - Z_3 + Z_4)\mathbb{I}[A \in \mathcal{G}_4] + (1.5 + 1.5Z_1 + 0.5Z_2 + Z_3 + 0.5Z_4)\mathbb{I}[A \in \mathcal{G}_5]\big)$.

As the number of treatments $M$ increases from 10, 15 to 20, the above three scenarios correspond to the treatments having 2, 3 and 5 treatment groups respectively. For each scenario, the true decision

boundaries are linear under the group domain. Take Scenario 1 for example. For any $\boldsymbol{z} \in \mathbb{R}^{10}$, the optimal ITR $D^*(\boldsymbol{z})$ would recommend any treatment in $\mathcal{G}_1$ when $0.2 - \boldsymbol{z}_1 - \boldsymbol{z}_2 < 0$, and any treatment in $\mathcal{G}_2$ when $0.2 - \boldsymbol{z}_1 - \boldsymbol{z}_2 \geqslant 0$. The treatments in all the scenarios are assigned in the unbalanced structure, which means the propensity score of specific treatments can be small. The tuning parameter $\lambda_n$ is selected by the 5-fold cross-validation. For each scenario, the training sample sizes vary from 200, 400 to 600 and we replicate the simulations for 200 times. On an independently generated testing data of size $10,000$, we evaluate the above methods using (a) empirical value function; (b) group-based misclassification rate between the estimated group decision rules and the optimal group decision rules. The empirical value function is calculated using $\mathbb{E}_n\big[\frac{\mathbb{I}(\widehat{D}(Z)=A)}{p(A|Z)}Y\big]/\mathbb{E}_n\big[\frac{\mathbb{I}(\widehat{D}(Z)=A)}{p(A|Z)}\big]$ [33, 10], where $\mathbb{E}_n$ denotes the empirical mean for the testing data. Since the treatment effects are identical within the same treatment group, the misclassification rate under the group domain is equivalent to that under the individual treatment domain.

**Results for simulation.** The boxplots of the empirical value and the misclassification rate evaluated on the independent testing data are shown in Figure 1 and Figure C.1 in the supplementary materials. In most cases, compared with RF, AD and PLS, our proposed SCAF method has superior performance with higher empirical values, smaller misclassification rates, and extremely lower variabilities for both evaluation criteria. When the treatment space becomes large, RF, AD and PLS suffer from inaccurate estimation of the individual treatment effect due to the small amount of observations for some specific treatments. Moreover, these methods do not consider the possible group structure in the treatment space and hence employ relatively large variance. As a comparison, SCAF is able to recover the treatment group structure and reduce the dimension of treatment space by clustering the treatments with similar treatment effects into the same treatment group. We also show the ratio of exactly recovering the true treatment group structure of SCAF among the 200 replications in Figure C.2 in the supplementary materials. Furthermore, we use Scenario 1 to demonstrate the advantage of our group-lasso step for finding the heterogeneous variables $X$ from $Z$. By implementing the group-lasso before the fusion step, as shown in Figure C.3 in the supplementary materials, SCAF with the group-lasso step has better performance than that without the group-lasso step since we only need to fuse the lower dimensional subvectors $\boldsymbol{\beta}_a$'s rather than $\boldsymbol{\zeta}_a$'s.

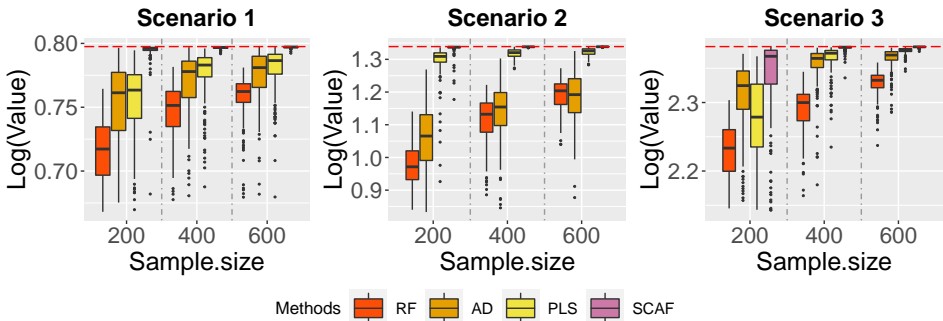

Figure 1: Boxplots of empirical value based on the testing data in simulations. Red dashed lines demonstrate the oracle values.

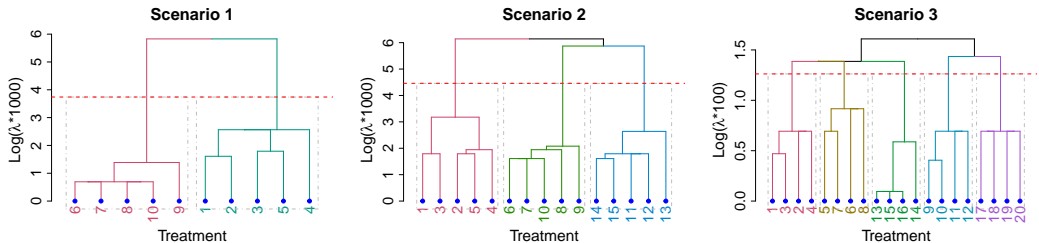

Figure 2: Solution path of the estimated treatment group structure as $\lambda$ increases. The true treatment group memberships are demonstrated with different colors. The red dotted horizontal lines show the best tuned $\lambda$ using cross-validation. The estimated groups based on the best $\lambda$ are framed with grey dotted rectangles.

Figure 2 demonstrates the solution path of the estimated treatment group structure as the tuning parameter $\lambda$ increases based on one of the replications for the three scenarios. When $\lambda = 0$, SCAF is equivalent to RF and PLS since the fusion penalty is not imposed and does not show any clustering pattern. As $\lambda$ increases, the fusion penalty encourages the treatments to merge together and cluster them with the expected group structure. When $\lambda$ becomes large enough, all of the treatments will be merged together and only one treatment group will be formed. With the best tuned $\lambda$, $2, 3$ and $5$ groups are correctly selected for the three scenarios respectively.

**Application to Patient-Derived Xenograft (PDX) study.** Due to the complexity of human cancer, there's significant heterogeneity of treatment effects among the high throughput patients' genomic biomarkers such as RNA and DNA sequencing [41]. Utilizing the transferred tumor pieces from patients to mice, the PDX study [42, 43, 44] aims to personalize the optimal cancer treatment for five types of cancer among a large number of FDA-approved preclinical cancer therapies, given the assigned treatments, the observed genomic features and responses. In particular, among five types of cancer, we focus on the ColoRectal Cancer (CRC), which includes $43$ PDX lines with complete genomic information. For CRC, 847 mice expanded from different PDX lines were treated with unique treatments among $M = 20$ single or combination treatments. The response $Y$ is measured by the scaled maximum observed tumor shrinkage from the baseline time, where the larger value is preferable. After the preprocessing and supervised screening steps shown in [29], we select 93 significant genomic biomarkers among all features.

Our goal is to identify the possible homogeneous group structure among these 20 treatments and cluster the treatments with similar treatment effects in order to boost the performance of the estimated ITR. The boxplot of the observed response for the 20 treatments in PDX data is shown in Figure 3. We can see that, with the highest mean and median of the response, the combination treatment BYL+BIN is superior than other treatments. For all methods, we randomly split the data into six folds with 200 replications where five folds are used to train the model in each replication. We

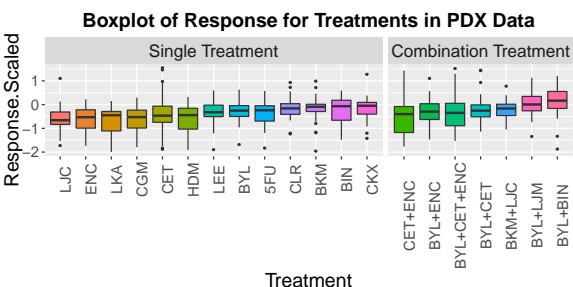

Figure 3: Observed response in PDX study.

use the remaining fold to compute the empirical value based on the estimated ITR. The tuning parameters are selected based on the three-fold cross validation.

**Results for the PDX study.** We first include all the single and combination treatments in the model. As shown in the left panel of Figure 4, with the highest empirical value, SCAF can provide more effective ITR than other comparison methods. On the overall dataset, we plot the solution path of the hierarchical clustering for all treatments in the left panel of Figure 5. With the best tuned $\lambda$, it can be seen that the 20 treatments can be clustered into three groups. Most of the combination treatments belong to the same group (the green one). One possible explanation is that they include BYL as a common component. It is interesting to point out that the combination treatment BYL+BIN forms a treatment group itself. Moreover, SCAF recommends BYL+BIN as the optimal treatment to $92.1\%$ patients among the PDX data, which is consistent with the outstanding performance of BYL+BIN shown in Figure 3. This also explains why BYL+BIN itself forms a group because its recommendation proportion nearly dominates the other treatments.

To better examine the group structure of the treatment effect, we exclude the combination treatments and only consider the 13 single treatments due to the superior performance of BYL+BIN. Based on the right panel of Figure 4, the overall values of all four methods decrease compared with the left panel because we drop the combination treatments and reduce the treatment space, while our proposed SCAF still has the best performance. Similarly, we draw the solution path of the hierarchical clustering for the single treatments in the right panel of Figure 5. The best tuned $\lambda$ further suggests that the single treatments can be clustered into three groups where the second treatment group (the green one) is recommended as the optimal to $94.1\%$ patients in the PDX data. The outstanding performance of the second group matches with Figure 3 because the treatments HDM and LKA in the first group (the pink one) and CGM, ENC and LJC in the third group (the blue one) are the treatments with smallest mean but highest variance among the 13 single treatments. In particular,

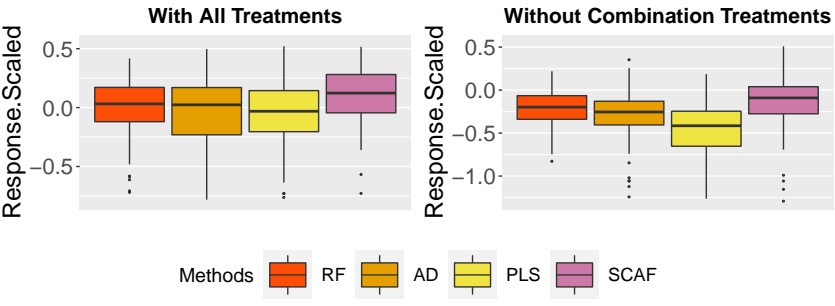

Figure 4: Boxplots of value on the testing data in PDX study.

Figure C.4 in the supplementary materials shows the path of empirical value function of the testing data for both cases. As $\lambda$ becomes larger, the empirical value reaches the maximum point due to the helpful clustering structure and finally dramatically decreases because of the excessive merging pattern imposed by the large $\lambda$. The results in Figure C.4 are consistent with Figure 5.

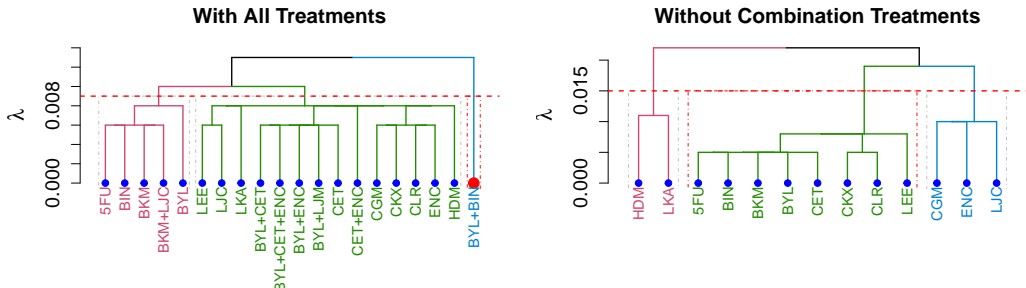

Figure 5: Solution paths for the PDX study as shown in Figure 2.

## 6    Discussion

In this paper, we consider the challenging ITR problem with a large number of treatments which may have homogeneous group structure. We propose the SCAF method to identify this structure and cluster the treatments in order to learn the optimal ITR more effectively. In particular, we adapt the idea of supervised clustering and formulate the problem as convex optimization that consists of *loss+penalty*. We are able to cluster the treatments and estimate the optimal ITR simultaneously within a single optimization problem. The whole clustering process can be intuitively visualized with a dendrogram plot similar to the hierarchical clustering. An efficient algorithm is proposed to solve the problem and the numerical results demonstrate the superior performance of SCAF.

The proposed method can be extended in several aspects. First, we consider the continuous outcome in this paper. One interesting future direction is to extend SCAF to deal with various types of outcomes, such as discrete outcome and survival outcome. Second, our method can be extended to learn the group structures of treatments for multi-stage dynamic treatment regimes [6, 11]. We leave these interesting directions for future research.

## Acknowledgments and Disclosure of Funding

The authors would like to thank the helpful and constructive comments from the reviewers which led to a much improved presentation of this paper. Yufeng Liu was supported in part by NSF grant DMS 2100729 and NIH grant R01GM126550. Donglin Zeng was partially supported by NIH R01GM124104, R01MH123487 and R01NS073671. Any opinions, findings, and conclusions or recommendations expressed in this material are those of the authors and do not necessarily reflect the views of National Science Foundation or National Institute of Health.

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
