# Supplement to "Learning Individualized Treatment Rules with Many Treatments: A Supervised Clustering Approach Using Adaptive Fusion"

**Haixu Ma**
Department of Statistics and Operations Research
University of North Carolina at Chapel Hill
Chapel Hill, NC 27516
haixuma@live.unc.edu

**Donglin Zeng**
Department of Biostatistics
University of North Carolina at Chapel Hill
Chapel Hill, NC 27516
dzeng@email.unc.edu

**Yufeng Liu**
Department of Statistics and Operations Research
Department of Genetics
Department of Biostatistics
University of North Carolina at Chapel Hill
Chapel Hill, NC 27516
yfliu@email.unc.edu

In this supplementary material, we present additional implementation details for the algorithm, proof of theorems, and additional figures for simulations and real data analysis.

## A  Additional details for the algorithm

### A.1  Estimation of the main effect

We briefly discuss how to obtain the estimation of the main effect function $M_0(Z)$ based on the weighted parametric regression or nonparametric regression models. By the identification condition in model (1), we have

$$M_0(Z) = \frac{\sum_{a=1}^{M} \mathbb{E}[Y|Z, A = a]}{M} = \mathbb{E}\big[\frac{Y}{Mp(A|Z)}|Z\big].$$

For parametric models, we assume the linear main effect $M_0(Z) = Z^{\mathsf{T}}\boldsymbol{\eta}$ where $\boldsymbol{\eta} \in \mathbb{R}^p$. Then, similar to [1] and [2], $\boldsymbol{\eta}$ can be estimated by the following $\ell_1$-penalized inverse-probability weighted regression problem:

$$\min_{\boldsymbol{\eta}} \left\{ \mathbb{E}_n\Big[ \Big(\frac{Y}{Mp(A|Z)} - Z^{\mathsf{T}}\boldsymbol{\eta}\Big)^2 \Big] + \lambda_{M_0} \|\boldsymbol{\eta}\|_1 \right\},$$

where the tuning parameter $\lambda_{M_0}$ can be selected using cross validation.

For nonparametric regression, we follow [3] to divide the training data into $M$ folds based on the assigned treatment. Then $\widehat{\mathbb{E}}[Y|Z, A = a]$ is obtained from the regression forest [4] on $Y \sim Z$ with the dataset $\{(y_i, \boldsymbol{z}_i) : a_i = a\}$. Finally, $\widehat{M}_0(Z) = \sum_{a \in \mathcal{A}} \widehat{\mathbb{E}}[Y|X, A = a]/M$. We refer to [3] for more discussions about the case of misspecifying the main effect, and the corresponding robust and efficient method to solve the misspecification problem.

36th Conference on Neural Information Processing Systems (NeurIPS 2022).

## A.2 Implementation details for the adaptive proximal gradient algorithm

Recall that $\mathbf{U} = \mathrm{diag}(\mathbf{X}_1, \mathbf{X}_2, \ldots, \mathbf{X}_M) \in \mathbb{R}^{n \times Mp}$ where $\mathbf{X}_a \in \mathbb{R}^{n_a \times p}$ is the submatrix of $\mathbf{X}$ and the observations in $\mathbf{X}_a$ are assigned to treatment $a$. Then we can rewrite $L_n(\boldsymbol{\beta}) = \frac{1}{2}\|\mathbf{U}\boldsymbol{\beta} - \bar{y}\|_2^2$ where $\bar{y} \in \mathbb{R}^n$ is the vector of calculated residual. The gradient of $L_n(\boldsymbol{\beta})$ can be directed calculated by $\nabla L_n(\boldsymbol{\beta}) = \mathbf{U}^\intercal(\mathbf{U}^\intercal\boldsymbol{\beta} - \bar{y})$ with Lipschitz constant $l_n = \lambda_{\max}(\mathbf{U}^\intercal\mathbf{U})$ where $\lambda_{\max}(\mathbf{U}^\intercal\mathbf{U})$ is the maximum eigenvalue of $\mathbf{U}^\intercal\mathbf{U}$. In addition, we follow [5] to approximately calculate the proximal operator of $P_n$ by solving the dual problem of $\mathrm{prox}_{s_n P_n}(\boldsymbol{\beta}) := \arg\min_{\bar{\boldsymbol{\beta}}} \left\{ P_n(\bar{\boldsymbol{\beta}}) + \frac{1}{2s_n}\|\bar{\boldsymbol{\beta}} - \boldsymbol{\beta}\|_2^2 \right\}$ for any updated $\boldsymbol{\beta}$ and the step size $s_n > 0$, with the accelerated projected gradient algorithm.

We use $\widehat{\boldsymbol{\beta}}^{(i)}$ to denote the estimation of $\boldsymbol{\beta}$ in the $i$-th iteration. Due to the usage of proximal gradient descent algorithm, the time and space complexities for our algorithm are both $\mathcal{O}(n^2)$, where $n$ is the training sample size. The main steps of the proposed algorithm for SCAF are summarized as below. In particular, the experiments were run on a Linux-based computing server.

---

**Algorithm 1: SCAF**

---

**Step 1**: Sort the observations based on the assigned treatment order.
**Step 2**: Remove the main effect $M_0(Z)$ and get residual $\bar{y}$.
**Step 3**: Implement group lasso to identify heterogeneous variables $X$ from $Z$.
**Step 4**: Use adaptive fast proximal gradient algorithm to solve problem (6) of the main paper:
  (1) Obtain the initial point $\boldsymbol{\beta}^{(0)}$ from Step 2 and set the desired tolerance $\epsilon_0 > 0$;
  (2) Compute the Lipschitz constant $l_n = \lambda_{\max}(\mathbf{U}^\intercal\mathbf{U})$ and set the step-size $s_n = 1/l_n$, $t_0 = 1$;
  (3) Let $\widehat{\boldsymbol{\beta}}^{(0)} := \boldsymbol{\beta}^{(0)}$ and set $\boldsymbol{\omega}_{l,t}^{(0)} := \min\{B_{\boldsymbol{\omega}}, 1/\|\widehat{\boldsymbol{\beta}}_l^{(0)} - \widehat{\boldsymbol{\beta}}_t^{(0)}\|_1\}$ for $P_n^{(0)}(\boldsymbol{\beta})$ $(l, t \in \mathcal{A})$;
  (4) For $i = 0, 1, \ldots, i_{\max}$, do:
    a. Compute $\boldsymbol{\beta}^{(i+1)} \approx \mathrm{prox}_{s_n P_n^{(i)}}\left(\widehat{\boldsymbol{\beta}}^{(i)} - s_n \nabla L_n(\widehat{\boldsymbol{\beta}}^{(i)})\right)$ [5];
    b. Update $t_{i+1} := (1 + \sqrt{1 + 4t_i^2})/2$;
    c. Perform FISTA [6] with $\widehat{\boldsymbol{\beta}}^{(i+1)} := \boldsymbol{\beta}^{(i+1)} + \frac{t_i - 1}{t_{i+1}}(\boldsymbol{\beta}^{(i+1)} - \boldsymbol{\beta}^{(i)})$;
    d. If $\|\widehat{\boldsymbol{\beta}}^{(i+1)} - \widehat{\boldsymbol{\beta}}^{(i)}\| \leqslant \epsilon_0$, then end the loop;
    e. Update $\boldsymbol{\omega}_{l,t}^{(i+1)} := \min\{B_{\boldsymbol{\omega}}, 1/\|\widehat{\boldsymbol{\beta}}_l^{(i+1)} - \widehat{\boldsymbol{\beta}}_t^{(i+1)}\|_1\}$ for $P_n^{(i+1)}(\boldsymbol{\beta})$ $(l, t \in \mathcal{A})$;
  (5) End of the main loop.
**Step 5**: Obtain the estimated ITR $\widehat{D}(\boldsymbol{x}) \in \arg\max_{a \in \mathcal{A}} \boldsymbol{x}^\intercal \widehat{\boldsymbol{\beta}}_a$ for $\boldsymbol{x} \in \mathcal{X}$.

---

# B Proof of theorems

## B.1 Proof of Theorem 1

Note that under the true group structure, we have

$$\bar{y} = \mathbf{H}\boldsymbol{\alpha}^0 + \boldsymbol{\epsilon}.$$

Since $\widehat{\boldsymbol{\alpha}}^{or} = (\mathbf{H}^{\intercal}\mathbf{H})^{-1}\mathbf{H}^{\intercal}\bar{y}$, we have

$$\widehat{\boldsymbol{\alpha}}^{or} - \boldsymbol{\alpha^0} = (\mathbf{H}^{\intercal}\mathbf{H})^{-1}\mathbf{H}^{\intercal}\boldsymbol{\epsilon}.$$

So,

$$\left\|\widehat{\boldsymbol{\alpha}}^{or} - \boldsymbol{\alpha^0}\right\|_{\infty} \leqslant \left\|(\mathbf{H}^{\intercal}\mathbf{H})^{-1}\right\|_{\infty}\left\|\mathbf{H}^{\intercal}\boldsymbol{\epsilon}\right\|_{\infty}.$$

We will bound $\left\|(\mathbf{H}^{\intercal}\mathbf{H})^{-1}\right\|_{\infty}$ and $\|\mathbf{H}^{\intercal}\boldsymbol{\epsilon}\|_{\infty}$ respectively.

First,

$$\begin{aligned}
\left\|(\mathbf{H}^{\intercal}\mathbf{H})^{-1}\right\|_{2} &= \sqrt{\lambda_{\max}^2\left((\mathbf{H}^{\intercal}\mathbf{H})^{-1}\right)} \\
&= \frac{1}{\lambda_{\min}\left(\mathbf{H}^{\intercal}\mathbf{H}\right)} \\
&\leqslant C_1^{-1}N_{\min}^{-1},
\end{aligned}$$

where the inequality is given by Assumption 2. Hence, we have

$$\left\|(\mathbf{H}^{\intercal}\mathbf{H})^{-1}\right\|_{\infty} \leqslant \sqrt{K_n p_n}\left\|(\mathbf{H}^{\intercal}\mathbf{H})^{-1}\right\|_{2} \leqslant \sqrt{K_n p_n}C_1^{-1}N_{\min}^{-1}.$$

Second, for $\|\mathbf{H}^{\intercal}\boldsymbol{\epsilon}\|_{\infty}$, denote $\boldsymbol{H}_j$ as the $j$-th column of $\mathbf{H}$. We have

$$\begin{aligned}
Pr\left(\|\mathbf{H}^{\intercal}\boldsymbol{\epsilon}\|_{\infty} > C\sqrt{n\log n}\right) &\leqslant \sum_{j=1}^{K_n p_n} Pr(|\boldsymbol{H}_j^{\intercal}\boldsymbol{\epsilon}| > C\sqrt{n\log n}) \\
&\leqslant \sum_{j=1}^{K_n p_n} Pr(|\boldsymbol{H}_j^{\intercal}\boldsymbol{\epsilon}| > C\|\boldsymbol{H}_j\|_2\sqrt{\log n}) \\
&\leqslant 2K_n p_n \exp(-c_1 C^2 \log n) = 2K_n p_n n^{-c_1 C^2},
\end{aligned}$$

where the second and third inequalities come from $\|\boldsymbol{H}_j\|_2 \leqslant \sqrt{n}$, Assumptions 1 and 3.

Combining both parts and let $C = c_1^{-1/2}$ complete the proof.

□

## B.2 Proof of Theorem 2

We follow the proof framework of [7]. Denote $\mathcal{M}_{\mathcal{G}} \subset \mathbb{R}^{M_n p_n}$ to be parameter space that has true group structure, i.e., $\mathcal{M}_{\mathcal{G}} = \left\{\boldsymbol{\beta} \in \mathbb{R}^{M_n p_n}, \text{ s.t., } \boldsymbol{\beta}_i = \boldsymbol{\beta}_j \text{ for } i, j \in \mathcal{G}_k, 1 \leqslant k \leqslant K\right\}$. Define the following two operators. (a) $T : \mathcal{M}_{\mathcal{G}} \to \mathbb{R}^{K_n p_n}$ and $T(\boldsymbol{\beta})$ is the $K_n p_n$-dimensional vector whose $k$-th $p_n$-dimensional vector is the common value of $\boldsymbol{\beta}_i$ for $i \in \mathcal{G}_k$. (b) $T^* : \mathbb{R}^{M_n p_n} \to \mathbb{R}^{K_n p_n}$ and

$$T^*(\boldsymbol{\beta}) = \left\{\frac{\sum_{i \in \mathcal{G}_k}\boldsymbol{\beta}_i}{|\mathcal{G}_k|}\right\}_{k=1}^{K_n}.$$

In particular, the operator $T$ will extract the distinct values of $\boldsymbol{\beta} \in \mathcal{M}_{\mathcal{G}}$. For any given vector $\boldsymbol{\beta} \in \mathbb{R}^{M_n p_n}$, the operator $T^*$ will construct a corresponding vector $T^*(\boldsymbol{\beta})$ that belongs to $\in \mathcal{M}_{\mathcal{G}}$ by taking the averaging value among the treatments within the same group. Then we can check that for $\boldsymbol{\beta} \in \mathcal{M}_{\mathcal{G}}, T(\boldsymbol{\beta}) = T^*(\boldsymbol{\beta})$. For any $\boldsymbol{\beta} \in \mathbb{R}^{M_n p_n}$, denote $\boldsymbol{\beta}^* = T^{-1}T^*(\boldsymbol{\beta}) \in \mathbb{R}^{M_n p_n}$ to be the vector expanded from $T^*(\boldsymbol{\beta})$ according to the true group structure.

Consider the following neighborhood of $\boldsymbol{\beta}^0$:

$$\Theta_n = \left\{\boldsymbol{\beta}, \text{ s.t., } \left\|\boldsymbol{\beta} - \boldsymbol{\beta}^0\right\|_{\infty} \leqslant \phi_n\right\},$$

where $\phi_n$ is defined in Theorem 1. From Theorem 1, we know that there exists an event $E_1$ where $Pr(E_1) \geqslant 1 - 2K_n p_n/n$, such that, conditional on $E_1$, we have $\widehat{\boldsymbol{\beta}}^{or} \in \Theta_n$. Now we aim to prove the following two arguments.

(1) For any $\boldsymbol{\beta} \in \Theta_n$ such that $\boldsymbol{\beta}^* \neq \widehat{\boldsymbol{\beta}}^{or}$, we have $Q_n(\boldsymbol{\beta}^*; \lambda_n) > Q_n(\widehat{\boldsymbol{\beta}}^{or}; \lambda_n)$.

(2) There exists another event $E_2$ where $Pr(E_2) \geqslant 1 - 2M_n p_n/n$, such that, conditional on the event $E_1 \cap E_2$, we have $Q_n(\boldsymbol{\beta}; \lambda_n) \geqslant Q_n(\boldsymbol{\beta}^*; \lambda_n)$ for any $\boldsymbol{\beta} \in \Theta_n$.

If (1) and (2) hold, then we have, for any $\boldsymbol{\beta} \in \Theta_n$, conditional on $E_1 \cap E_2$,

$$Q_n(\boldsymbol{\beta}; \lambda_n) \geqslant Q_n(\boldsymbol{\beta}^*; \lambda_n) > Q_n(\widehat{\boldsymbol{\beta}}^{or}; \lambda_n).$$

In other words, the oracle estimator $\widehat{\boldsymbol{\beta}}^{or}$ is the strictly local minimizer of $Q_n(\boldsymbol{\beta}; \lambda_n)$ in the neighborhood $\Theta_n$ with probability greater than $1 - 2(K_n p_n + M_n p_n)/n$ when $n$ is sufficiently large. Then the results follow.

Now, we start to prove (1) and (2).

Proof of (1): For any $\boldsymbol{\beta} \in \Theta_n$, denote $T^{-1}T^*(\boldsymbol{\beta}) = \boldsymbol{\beta}^* = (\boldsymbol{\beta}_1^*, \ldots, \boldsymbol{\beta}_{M_n}^*)^{\mathsf{T}} \in \mathcal{M}_{\mathcal{G}}$ and denote $T^*(\boldsymbol{\beta}) = \boldsymbol{\alpha} = (\boldsymbol{\alpha}_1, \cdots, \boldsymbol{\alpha}_{K_n})^{\mathsf{T}}$. Note that the oracle estimator is the unique minimizer of the $L_2$ loss, which is the first part of $Q_n(\boldsymbol{\beta}; \lambda_n)$. Hence, we can only prove that for any $\boldsymbol{\beta}^* \in \Theta_n \cap \mathcal{M}_{\mathcal{G}}$, the penalty term

$$\sum_{1 \leqslant l < t \leqslant M_n} p_{\lambda_n}(\|\boldsymbol{\beta}_l^* - \boldsymbol{\beta}_t^*\|_1) = \sum_{1 \leqslant k < k' \leqslant K_n} |\mathcal{G}_k| |\mathcal{G}_{k'}| p_{\lambda_n}(\|\boldsymbol{\alpha}_k - \boldsymbol{\alpha}_{k'}\|_1),$$

is a constant. To prove that, based on Assumption 4, we can only show that $\|\boldsymbol{\alpha}_k - \boldsymbol{\alpha}_{k'}\|_1 \geqslant \frac{a}{2}\lambda_n$ for any $k \neq k'$. Note that

$$\begin{aligned}
\|\boldsymbol{\alpha}_k - \boldsymbol{\alpha}_{k'}\|_1 \geqslant \|\boldsymbol{\alpha}_k - \boldsymbol{\alpha}_{k'}\|_\infty &\geqslant \|\boldsymbol{\alpha}_k^0 - \boldsymbol{\alpha}_{k'}^0\|_\infty - 2\|\boldsymbol{\alpha} - \boldsymbol{\alpha}^0\|_\infty \\
&= \|\boldsymbol{\alpha}_k^0 - \boldsymbol{\alpha}_{k'}^0\|_\infty - 2 \sup_{1 \leqslant k \leqslant K_n} \left\| \sum_{i \in \mathcal{G}_k} \frac{\boldsymbol{\beta}_i - \boldsymbol{\beta}_i^0}{|\mathcal{G}_k|} \right\|_\infty \\
&\geqslant \|\boldsymbol{\alpha}_k^0 - \boldsymbol{\alpha}_{k'}^0\|_\infty - 2 \sup_{1 \leqslant k \leqslant K_n} \sup_{i \in \mathcal{G}_k} \|\boldsymbol{\beta}_i - \boldsymbol{\beta}_i^0\|_\infty \\
&\geqslant \|\boldsymbol{\alpha}_k^0 - \boldsymbol{\alpha}_{k'}^0\|_\infty - 2\|\boldsymbol{\beta} - \boldsymbol{\beta}^0\|_\infty \\
&\geqslant b_n/p_n - 2\phi_n \geqslant a\lambda_n - 2\phi_n \gg \frac{a}{2}\lambda_n \quad \text{(By Assumption 5)}.
\end{aligned}$$

Hence, the result follows.

Proof of (2): Recall the definition of $L_n(\boldsymbol{\beta})$ in (7) and recall that

$$\mathbf{U} = \begin{pmatrix} \mathbf{X}_1 & & & \\ & \mathbf{X}_2 & & \\ & & \ddots & \\ & & & \mathbf{X}_M \end{pmatrix}_{n \times Mp}.$$

For any $\boldsymbol{\beta} \in \Theta_n$, we have

$$Q_n(\boldsymbol{\beta}; \lambda_n) - Q_n(\boldsymbol{\beta}^*; \lambda_n) = \underbrace{L_n(\boldsymbol{\beta}) - L_n(\boldsymbol{\beta}^*)}_{\Gamma_1} + \underbrace{\sum_{1 \leqslant l < t \leqslant M_n} p_{\lambda_n}(\|\boldsymbol{\beta}_l - \boldsymbol{\beta}_t\|_1) - \sum_{1 \leqslant l < t \leqslant M_n} p_{\lambda_n}(\|\boldsymbol{\beta}_l^* - \boldsymbol{\beta}_t^*\|_1)}_{\Gamma_2}.$$

By Taylor expansion,

$$\Gamma_1 = -\left[\mathbf{U}^{\mathsf{T}}\bar{y} - \mathbf{U}^{\mathsf{T}}\mathbf{U}\bar{\boldsymbol{\beta}}\right]^{\mathsf{T}}(\boldsymbol{\beta} - \boldsymbol{\beta}^*),$$

where $\bar{\boldsymbol{\beta}} = \xi\boldsymbol{\beta} + (1 - \xi)\boldsymbol{\beta}^*$ and $\xi \in (0, 1)$. For the gradient part, let

$$\boldsymbol{w} = (\boldsymbol{w}_1, \boldsymbol{w}_2, \ldots, \boldsymbol{w}_{M_n})^{\mathsf{T}} := \mathbf{U}^{\mathsf{T}}\bar{y} - \mathbf{U}^{\mathsf{T}}\mathbf{U}\bar{\boldsymbol{\beta}},$$

where $\boldsymbol{w}_m \in \mathbb{R}^{p_n}$ for any $m = 1, \ldots, M_n$. Then

$$\Gamma_1 = -\boldsymbol{w}^{\mathsf{T}}(\boldsymbol{\beta} - \boldsymbol{\beta}^*)$$

$$= -\sum_{k=1}^{K_n} \sum_{i \in \mathcal{G}_k} \sum_{j \in \mathcal{G}_k} \frac{\boldsymbol{w}_i^\intercal (\boldsymbol{\beta}_i - \boldsymbol{\beta}_j)}{|\mathcal{G}_k|}$$

$$= -\sum_{k=1}^{K_n} \sum_{i \in \mathcal{G}_k} \sum_{j \in \mathcal{G}_k} \frac{(\boldsymbol{w}_j - \boldsymbol{w}_i)^\intercal (\boldsymbol{\beta}_j - \boldsymbol{\beta}_i)}{2|\mathcal{G}_k|}$$

$$= -\sum_{k=1}^{K_n} \sum_{i,j \in \mathcal{G}_k, i<j} \frac{(\boldsymbol{w}_j - \boldsymbol{w}_i)^\intercal (\boldsymbol{\beta}_j - \boldsymbol{\beta}_i)}{|\mathcal{G}_k|}$$

$$\geqslant -\sum_{k=1}^{K_n} \sum_{i,j \in \mathcal{G}_k, i<j} \frac{\|\boldsymbol{w}_j - \boldsymbol{w}_i\|_\infty \|\boldsymbol{\beta}_j - \boldsymbol{\beta}_i\|_1}{|\mathcal{G}_k|}$$

$$\geqslant -\sum_{k=1}^{K_n} \sum_{i,j \in \mathcal{G}_k, i<j} \frac{2\|\boldsymbol{w}\|_\infty \|\boldsymbol{\beta}_j - \boldsymbol{\beta}_i\|_1}{|\mathcal{G}_k|}.$$

Note that based on the definition of $\boldsymbol{w}$, we have

$$\|\boldsymbol{w}\|_\infty \leqslant \|\mathbf{U}^\intercal \mathbf{U} (\boldsymbol{\beta}_0 - \bar{\boldsymbol{\beta}})\|_\infty + \|\mathbf{U}^\intercal \boldsymbol{\epsilon}\|_\infty$$
$$\leqslant \mathcal{O}(K_n p_n \phi_n) + \|\mathbf{U}^\intercal \boldsymbol{\epsilon}\|_\infty.$$

Similar to the previous proof, we have $Pr\left(\|\mathbf{U}^\intercal \boldsymbol{\epsilon}\|_\infty \leqslant \mathcal{O}(\sqrt{n \log n})\right) \geqslant 1 - 2M_n p_n/n$. Hence, we have

$$\Gamma_1 \geqslant -\sum_{k=1}^{K_n} \sum_{i,j \in \mathcal{G}_k, i<j} \frac{\mathcal{O}(\sqrt{n \log n}) \|\boldsymbol{\beta}_j - \boldsymbol{\beta}_i\|_1}{|\mathcal{G}_k|},$$

with probability at least $1 - 2M_n p_n/n$.

For $\Gamma_2$, note that for treatments $i, j$ that belong to two different groups $\mathcal{G}_k$ and $\mathcal{G}_{k'}$, we have

$$\|\boldsymbol{\beta}_i - \boldsymbol{\beta}_j\|_1 \geqslant \|\boldsymbol{\beta}_i - \boldsymbol{\beta}_j\|_\infty \geqslant \|\boldsymbol{\beta}_i^0 - \boldsymbol{\beta}_j^0\|_\infty - 2\|\boldsymbol{\beta} - \boldsymbol{\beta}^0\|_\infty \geqslant b_n/p_n - 2\phi_n \geqslant a\lambda_n - 2\phi_n \gg \frac{a}{2}\lambda_n.$$

In addition, since $\boldsymbol{\beta} \in \Theta_n$, we have $\boldsymbol{\beta}^* \in \Theta_n$ as well. Hence, with similar derivations, we have $\|\boldsymbol{\beta}_i^* - \boldsymbol{\beta}_j^*\|_1 \gg \frac{a}{2}\lambda_n$. Based on Assumption 4,

$$\sum_{i \in \mathcal{G}_k, j \in \mathcal{G}_{k'}, k \neq k'} p_{\lambda_n}(\|\boldsymbol{\beta}_i - \boldsymbol{\beta}_j\|_1) - \sum_{i \in \mathcal{G}_k, j \in \mathcal{G}_{k'}, k \neq k'} p_{\lambda_n}(\|\boldsymbol{\beta}_i^* - \boldsymbol{\beta}_j^*\|_1) = 0$$

Therefore, only the treatments that belong to the same group contribute to $\Gamma_2$. According to the same calculation in the proof of Theorem 2 from [7], we have

$$\Gamma_2 = \sum_{k=1}^{K_n} \sum_{i,j \in \mathcal{G}_k, i<j} p_{\lambda_n}(\|\boldsymbol{\beta}_i - \boldsymbol{\beta}_j\|_1) - \sum_{k=1}^{K} \sum_{i,j \in \mathcal{G}_k, i<j} p_{\lambda_n}(\|\boldsymbol{\beta}_i^* - \boldsymbol{\beta}_j^*\|_1)$$
$$\geqslant \sum_{k=1}^{K_n} \sum_{i,j \in \mathcal{G}_k, i<j} p'_{\lambda_n}(\|\bar{\boldsymbol{\beta}}_i - \bar{\boldsymbol{\beta}}_j\|_1) \|\boldsymbol{\beta}_i - \boldsymbol{\beta}_j\|_1.$$

Combining the bound for $\Gamma_1$ and $\Gamma_2$, we have

$$Q_n(\boldsymbol{\beta}; \lambda_n) - Q_n(\boldsymbol{\beta}^*; \lambda_n) = \Gamma_1 + \Gamma_2$$
$$\geqslant \sum_{k=1}^{K_n} \sum_{i,j \in \mathcal{G}_k, i<j} \left( p'_{\lambda_n}(\|\bar{\boldsymbol{\beta}}_i - \bar{\boldsymbol{\beta}}_j\|_1) - \frac{\mathcal{O}(\sqrt{n \log n})}{|\mathcal{G}_k|} \right) \|\boldsymbol{\beta}_i - \boldsymbol{\beta}_j\|_1.$$

Note that $\|\bar{\boldsymbol{\beta}}_i - \bar{\boldsymbol{\beta}}_j\|_1 \leqslant \|\bar{\boldsymbol{\beta}}_i - \boldsymbol{\beta}_i^0\|_1 + \|\bar{\boldsymbol{\beta}}_j - \boldsymbol{\beta}_j^0\|_1 \leqslant 2\phi_n$. Hence, based on Assumption 4, $p'_{\lambda_n}(\|\bar{\boldsymbol{\beta}}_i - \bar{\boldsymbol{\beta}}_j\|_1) \geqslant \mathcal{O}(\sqrt{n \log n})/\inf_{1 \leqslant k \leqslant K_n} |\mathcal{G}_k|$. This completes the proof. □

## C  Additional Figures

## D  PDX Data

The PDX data we used in real data analysis can be downloaded from `https://www.tandfonline.com/doi/suppl/10.1080/01621459.2020.1828091?scroll=top`.

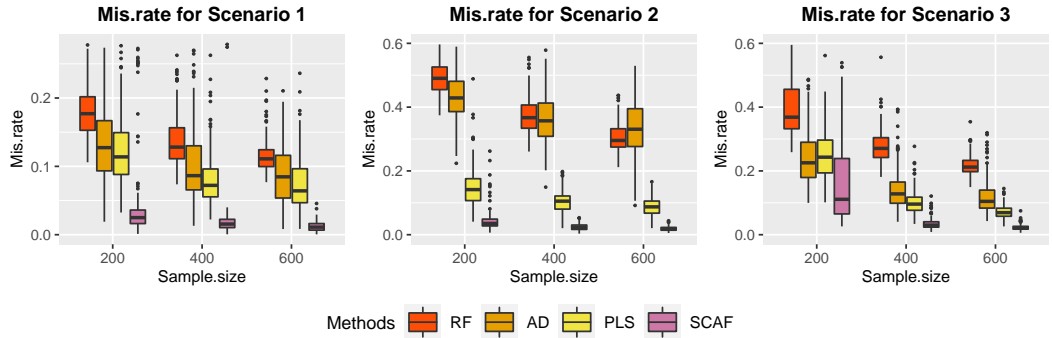

Figure C.1: Boxplots of misclassification rate based on the testing data in simulations.

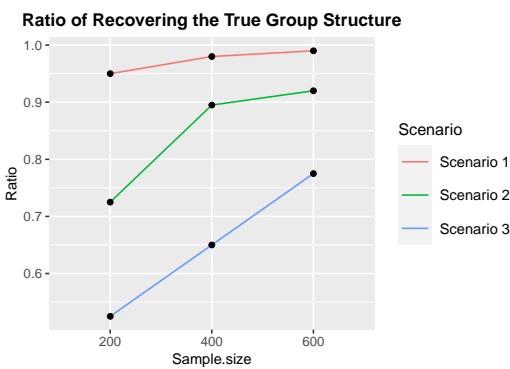

Figure C.2: Ratio of recovering the true group structure among 200 replications in simulations.

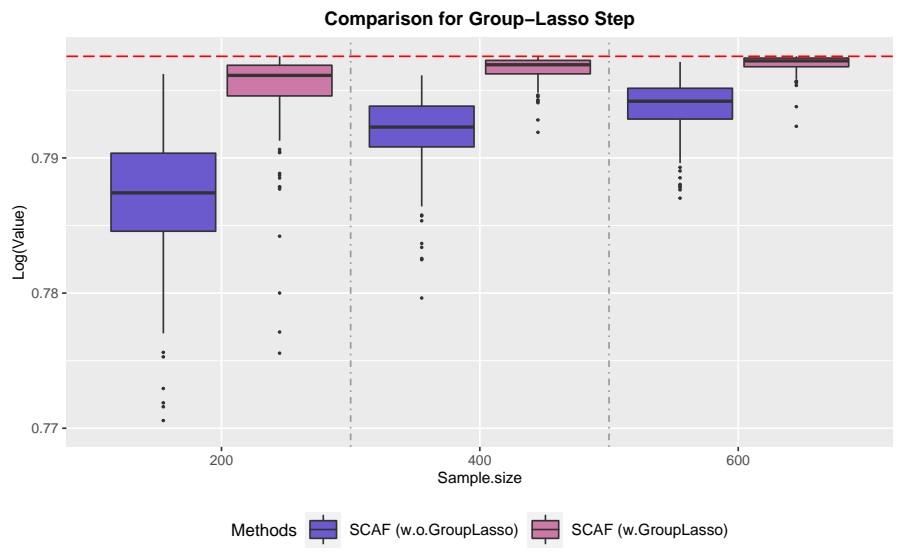

Figure C.3: Boxplots of empirical value for SCAF (with/without group-lasso step) in Scenario 1.

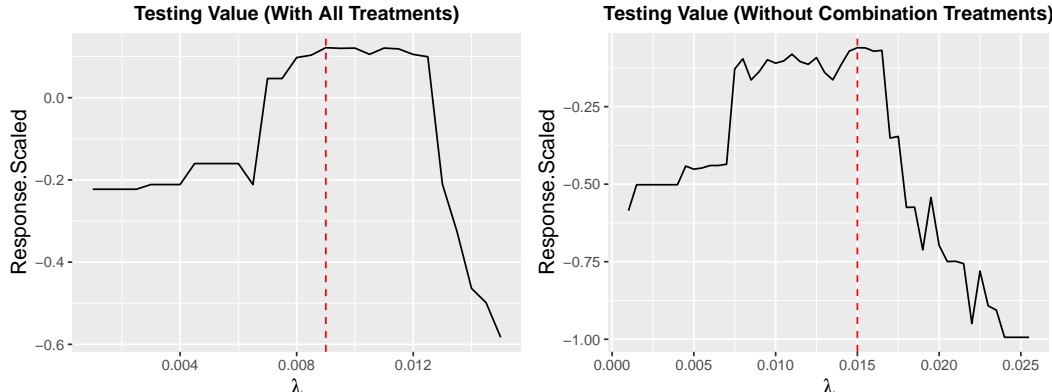

Figure C.4: Path of empirical value on the testing data as $\lambda$ increases in PDX study. The red vertical dotted lines show the best tuned $\lambda$ using cross-validation.