# OpenReview forum: "Learning Individualized Treatment Rules with Many Treatments: A Supervised Clustering Approach Using Adaptive Fusion"
_NeurIPS.cc/2022/Conference — NeurIPS 2022 Accept_

### Official Review · Reviewer_DGyw · 2022-06-21

**Rating:** 7
**Confidence:** 4
**Soundness:** 3 good
**Presentation:** 3 good
**Contribution:** 3 good

**Summary:**

This submission deals with treatment selection among a large collection of candidate ones for a specific individual or a group of individuals which share common response to the treatment. The authors propose to adopt a treatment clustering strategy by gathering treatments with similar effects.

The proposed approach follows the line of thought of some prior arts in which a statistical regression model is defined for the reward viewed as a function comprising a general and common effect incurred from all treatments plus some treatment specific effects and an additive noise. The common and treatment specific parts of the reward are functions of the patient whose information is embedded in a fixed-size vector. The goal of the authors is to learn the regression coefficients based on supervision coming from data in the form triplets (patient, treatment, reward). They leverage a group promoting penalty to encourage treatments with similar effects to be clustered together.
To rule out some of the covariates inside the vectors of patient features, the authors first use a group lasso (on the same reward regression) as a preliminary step. Finally, the main regression problem has properties allowing it to be solved by an accelerated proximal gradient algorithm.

The paper is technically sound and quite well written. A theorem proves the consistency of the estimator under technical (but not unusual) assumptions.

**Questions:**

Major :

Have the authors considered investigating more flexible models by using for instance a kernelized version of linear regression ?

I think that the interpretation of random variable $A$ should be clarified at the beginning of section 2. Indeed, should it be understood as the identity of the best treatment, or is it the result of some sampling distribution (e.g. uniform) from which the data is obtained ? Because it is difficult to imagine obtaining data from the former, I suspect the correct interpretation is the latter. The action sampling process in the first experiment of section 5 should be documented.

How is the term $M_0(.)$ estimated in the real data experiment in section 5 ?

Minor :

For better readability, I suggest not to use bold letters for scalar variables such as $\zeta_{a,k}$.

**Limitations:**

Not adapted to discrete rewards and limited learning capacity.

**Strengths And Weaknesses:**


Pros :
- the paper incorporates a novel aspect (treatment grouping) in a standard model used for personalized treatment selection
- the proposed method has a theoretical group-consistency property
- the efficiency of the approach is experimentally validated on synthetic and real data

Cons :
- the model is designed for continuous outputs whereas discrete ones may be observed for several treatments/pathologies
- the model backbone is a linear regression and may underfit the data in various situations

---

> ### Author Response · Authors · 2022-08-01
> **Reply to the comments**
>
> Thank you for the comprehensive summary and constructive comments. Please see the following response to your concerns.
>
> 1. Reply to your comment "The model is designed for continuous outputs whereas discrete ones may be observed for several treatments/pathologies.":
>
> Although our paper focuses on the continuous outcome case, we can extend our method to deal with discrete outcome (use generalized linear models) or survival outcome. We just need to replace the loss term accordingly and still keep the fusing penalty to achieve treatment clustering.
>
> 2. Reply to your comment "The model backbone is a linear regression and may underfit the data in various situations." and "Have the authors considered investigating more flexible models by using for instance a kernelized version of linear regression?":
>
> Our paper mainly deals with linear regression. The more flexible regression function can be generalized by adding the polynomial terms and implementing kernel regression as you suggested. We can still use fusing penalty in that case. We will explore more in the future.
>
> 3. Reply to your comment "I think that the interpretation of random variable $A$ should be clarified at the beginning of Section 2. Indeed, should it be understood as the identity of the best treatment, or is it the result of some sampling distribution (e.g. uniform) from which the data is obtained? Because it is difficult to imagine obtaining data from the former, I suspect the correct interpretation is the latter. The action sampling process in the first experiment of Section 5 should be documented.":
>
> For the clarification about $A$ we used in the paper, just as you indicated, $A$ is denoted as the assigned treatments from some sampling distribution. The recommended treatment is described by $D(X)$. We will further clarity this at the beginning of Section 2.
>
> 4. Reply to your comment "How is the term  $M_0(X)$ estimated in the real data experiment in Section 5?":
>
> For estimating the main effect $M_0(X)$, please refer to Section A.1 in the supplementary materials. We include detailed description for both parametric and non-parametric estimation methods.
>
> 5. Reply to your comment "For better readability, I suggest not to use bold letters for scalar variables such as $\zeta_{a,k}$.":
>
> Thanks for your suggestion about the terminology. We will revise the paper accordingly to improve the presentation.

---

### Official Review · Reviewer_BQsP · 2022-06-30

**Rating:** 6
**Confidence:** 3
**Soundness:** 3 good
**Presentation:** 3 good
**Contribution:** 3 good

**Summary:**

The work proposes a machine learning approach for precision medicine applications, where one needs to choose from a set of suitable treatments the best one for a given patient. Further, the proposed approach allows clustering similarly behaving treatments together, and the authors give theoretical guarantees that the underlying clustering will be recovered.

The proposed approach combines together a number of different approaches including:
- Basic regression model, where the outcome is modelled as a sum of a main effect M(z), and interaction effect T(z,a)
- After solving a standard regression problem to recover M(z), group Lasso is used to find the subset of features that may contribute to the interaction effect
- Proposed adaptive proximal gradient algorithm is used to predict the remaining residual with T(z,a), and a pairwise regularizer is used to divide the treatments into clusters

Experiments on simulated data show that the approach works on finding the true underlying structure on simple simulated problems, and evaluation on a cancer treatment data set shows both competitive performance against basic ML regression approaches, as well as the treatment structure recovered by the method.



**Questions:**

- How does the proposed method compare on real data to the baseline that always chooses the treatment that worked best on average on training data, and predicts the mean effect for that treatment? I find it a bit difficult to interpret from the metrics provided, how much is actually learned from the data compared to just predicting the mean.
- What does the "response scaled" -metric mean, what units, scaled how? Maybe something to add to the supplementary?
- Regarding comparison to the baseline methods, it would be good to tell in the supplementary materials the used parameter grids, and also what kind of feature representations these methods were supplied with. For example, if the goal is to learn both M(z) and T(z,a), in what form was the identity of the treatment a supplied to the baseline methods? The code is provided, but it would be good to be able to tell these details already from reading paper + supplementaries, in order to understand what the comparison means.

**Limitations:**

Limitations and negative societal impact not considered. The one possible limitation that comes to my mind, is that the method should be much more comprehensively validated on more realistic data sets and settings, before it could be ever considered for making treatment decisions for real-world patients.

**Strengths And Weaknesses:**

The paper addresses an important problem in precision medicine, and provides ideas that appear to be novel about how to, in addition to predicting, simultaneously cluster together similar treatments. The writing is fairly clear, the work and contributions well motivated, and I found the mathematical presentation to be rigorous yet accessible.

The proposed learning approach seems reasonable, though to me some of the choices made to make divide this into a sequence of solvable (convex) optimization problems feel a bit like not so well justified "hacks" (solving for M(z) and T(z,a) separately, the assumption that Z decomposes nicely to X and V that can be recovered with group Lasso...). Theoretical analysis seems convincing, but I cannot verify that all the details are correct.

The experimental analysis on the PDX study is interesting, though from results on a single data set it is difficult to ascertain yet how well the method can be in general expected to perform. The one thing that was left a bit unclear to me was does the proposed approach perform more accurate predictions, than the simple approach of doing non-personalized predictions (i.e. always predicting the one treatment that works best on training data, on the benchmark data most likely usually the BYL+BIN treatment). Based on zooming on Figures 3 and 4 this might be the case, but this could be more clearly discussed in the text.

Minor comments:

Equation (1): I was a bit surprised there was not in addition to M(z) and T(z,a) a U(a) type of term that would depend only on the treatment, since in addition to personalized effects one would except some of the treatments to be overall superior to others. E.g. the BYL+BIN treatment in the PDX study. Though I suppose such effect can be encoded into T(z,a) with constant covariate for z, and identity encoding covariate for a...

"In practice, one may assume that only certain elements of Z...[divide into X and V]" - this is not intuitively obvious to me, as in many cases one could imagine same covariates contributing both to M and Z? Any plausible real-world examples of this?

I did wonder if some of the reinforcement learning type of terminology used in the beginning (rewards, value functions) were necessary, since in the end the considered setup is regression where training data has been gathered in advance, and gathering feedback and considering exploration/exploitation types of tradeoffs is not considered. The presentation could be simpler by leaving these connections out, but perhaps there is also value in making these connections.

---

> ### Author Response · Authors · 2022-08-01
> **Reply to the comments**
>
> Thanks for your summary of the paper and constructive comments. Please see our following clarifications below.
>
> 1. For your concerns about dividing the problem into two convex optimization problems, using group lasso to identify $X$ and $V$ from $Z$ (first optimization problem) is helpful but is not necessary. Without this step, we can still implement our proposed fusion penalty (second optimization problem) to cluster the treatments. The group-lasso step is helpful to  save some computational time (fuse a lower dimensional vector) in the second step and may improve the performance of estimated ITR. Our methodology and theoretical contributions mainly focus on the fusion step that solves the ITR problem.
>
> 2. Thanks for your interesting comments on the PDX data analysis. When including all the treatments, recommending BYL+BYN, which has the largest treatment effect for $\textbf{most}$ patients, to $\textbf{all}$ patients gives a mean value of 0.121, compared with a mean value of 0.125 (larger the better) if individualized treatment recommendation is implemented. We will add this analysis in the paper. Thus, individualization is still helpful though not significant. Furthermore, our estimated treatment structure shows that the combination treatment BYL+BYN forms a group itself (see the left panel of Figure 5), which provides informative and consistent knowledge about the superior performance of BYL+BYN. However, BYL+BYN is a combination treatment of two medications, which could induce high cost and possible complicate side effects. That’s why we also exclude the combination treatments and implement our method for the single treatment set (see in the right panel of Figure 5) to find the “second” optimal individualized treatment recommendation. In this case, without the dominant performance of BYL+BYN, the individualized recommendation has significant better performance than any other non-personalized treatment rule.
>
> 3. For your comments about the possible $U(A)$ term in Equation (1), this treatment specific effect can be indeed combined into the intercept term in $T(Z, A)$. Our paper deals with this term exactly as you indicated.
>
> 4. Here is a clarification about your second minor comments. As long as the variable contributes to the interaction term $T$ (can contribute to both $T$ and $M$, or only contribute to $T$), it is considered as the heterogeneous variables $X$. If the variable only shows up in the main effect $M$, it belongs to the homogeneous variables $V$. However, as mentioned in our clarification shown in 1, the prior knowledge or the group lasso step about $X$ and $V$ is helpful but not required. In practice, we may exclude some sensitive characteristics from $X$ due to the consideration of fairness.
>
> 5. The terminology of rewards and value function are commonly used in the individualized decision-making literature and are not restricted in the reinforcement learning area. Please refer more details in the following key references in the ITR literature: (1) Qian and Murphy. (2012), Performance guarantees for individualized treatment rules, Annals of Statistics; (2) Zhao et al. (2012) Estimating individualized treatment rules using outcome weighted learning, Journal of the American Statistical Association; (3) Chen et al. (2020) Representation learning for integrating multi-domain outcomes to optimize individualized treatments, Neurips 2020.
>
> 6. For the evaluation metric we utilized in line 311, it is a typical evaluation criterion that is commonly used in the individualized decision-making area (see the above papers in 5 for more details about the interpretation). Compared to the possible misspecification and overfitting issue from predicting the mean effect of treatments, this metric is a nonparametric, unbiased and robust estimator of the value function.
>
> 7. Sorry for the confusing description about the “Response Scaled” in Figures 3 and 4. The response is defined in “Rashid et al. (2021), High-dimensional precision medicine from patient-derived xenografts, Journal of the American Statistical Association”. The defined response is specific in PDX analysis. It is related to the size of tumor. We just follow this definition and words in the above paper. We will clarify this in the supplements.
>
> 8. Thanks for the suggestion about the clear demonstration for the comparison methods. We will add this in the supplements.
>
> 9. We will try more real data and comprehensively validate our methods in the future.
>
> 10. For the social impact, if a certain patient group in the training data is under representative, we can reweight the samples to alleviate the fairness issue. We will add this in our discussion section.

---

> > ### Comment · Reviewer_BQsP · 2022-08-05
> > **Response to authors**
> >
> > Thank you for your comprehensive responses that helped me to better understand this work, as well as the proposed clarifications for the paper and the supplementary materials. Perhaps point 3 could be very briefly clarified also in the paper?

---

> > > ### Author Response · Authors · 2022-08-05
> > > **Reply to the comments**
> > >
> > > We appreciate your prompt response and acknowledgement of our responses. We are grateful for the increased score which is very encouraging. Following your suggestion, we will add more clarifications about the point 3 in our paper. As you pointed out, a possible treatment effect term $U(A)$ that only contributes from the treatments is often considered. This treatment-specific term can be combined into the intercept term in $T(Z, A)$ in our model. Therefore, our proposed model is flexible to deal with this term.

---

### Official Review · Reviewer_cwpD · 2022-07-07

**Rating:** 6
**Confidence:** 4
**Soundness:** 3 good
**Presentation:** 4 excellent
**Contribution:** 3 good

**Summary:**

The authors consider the problem of finding optimal individualised treatment
rules – an important problem in precision medicine – that assigns treatments based
on covariates which may change over time. The particular setting of interest
is one where many treatments exist and where treatments have a structure induced
by shared mechanisms of action (e.g., different drugs that target the same pathway).

The proposed approach uses a fusion penalty term that encourages clustering
between treatments. This incorporates the structure assumption in a tunable way,
and allows sharing of information between infrequently observed treatments and
more frequently observed ones that are related.


**Questions:**

- The PDX study employes a number of preprocessing steps, some of which are
  supervised. How was bias avoided with the supervised preprocessing? Was
  separate data used?

- The treatments employed are surely employing drugs with known targets. Do the
  targets/targeted pathways align with the inferred grouping structure?


**Limitations:**

- Given the strong patient focus in the intro, the confounding limitation should
  be discussed. The PDX study used in experiments is free of confounding, but
  typical observational patient data is not.


**Strengths And Weaknesses:**

- The literature is well covered in their citations, with the exception of
  confounding. There are works considering ITRs in the context of confounding
  through the use of instrumental variables. This is not strictly a problem
  since the authors aren't considering confounding.

- The work builds on existing with the novelty being the fusion term.


- The formulation is straightforward and easy to understand, as is the writing
  and presentation.

- There are two parts to the method, first there's the group lasso based
  classification into homogeneous and heterogenous variables, and second is the
  optimisation of the main objective function with the fusion penalty term. Most
  of the work concentrates on the second part: the theoretical properties assume
  the hetrogeneous variables are given and the experiments do not specifically
  investigate the identification of homo/hetrogenous variables. There is a short
  note on lines 150–152 stating it has been empirically observed to be more
  effective, but these are not shown anywhere. Furthermore, it's unclear if the
  group lasso classification was used in experiments or not, and if it was how
  the hyperparameter was chosen.

- The theoretical results are relevant and interesting but are limited to the
  fusion model only.

- The PDX study is not sufficiently described, it is unclear exactly which
  dataset is used as no reference is provided. There is a citation to another
  methods paper, cited as the authors have used their preprocessing steps, but
  no citation to the main data reference.

---

> ### Author Response · Authors · 2022-08-01
> **Reply to the comments**
>
> Thanks for your nice summary of the paper and constructive comments. Please see our clarifications below.
>
> 1. Thanks for pointing out the confounding issue in the ITR problem. Indeed, the confounding issue is very important in observational studies. The data generalization and the real data (PDX study) we considered in the paper mainly focus on the clinical trial setting. Thus, the confounding issue is not a major concern. For observational studies, our method can be further generalized to deal with confounding under certain assumptions, i.e., using propensity scores. It is an interesting future direction to explore. We will add some brief discussion in the paper.
>
> 2. For the first step of our algorithm (use group lasso to identify heterogeneous variables), it is helpful but is not required. Without this step, we can still implement our proposed fusion penalty to cluster the treatments. The group-lasso step is helpful to save computational time (fuse a lower dimensional vector) in the second step and may improve the performance of estimated ITR. Our methodology and theoretical contributions mainly focus on the fusion step.
>
> 3. The group lasso step is implemented in our empirical study and the tuning parameters are selected using cross validation. It was shown that it can boost the performance of our estimated ITR. We did not add the comparison results in the main paper due to the page limitation. We will further add that in the supplement.
>
> 4. Thanks for the comments about the detailed description of the PDX data. More introduction and background about PDX study can be seen in “Gao et al. (2015), High-Throughput Screening Using Patient-Derived Tumor Xenografts to Predict Clinical Trial Drug Response, Nature Medicine”. We will add this reference in our main paper.
>
> 5. We did not use the separate data, and we followed the same preprocessing steps shown in “Rashid et al. (2021), High-dimensional precision medicine from patient-derived xenografts, Journal of the American Statistical Association”. The preprocessing steps are shown to have satisfactory performance in this paper. We will try other preprocessing steps to explore whether we can improve the results.
>
> 6. For your question about the alignment between the targets/targeted pathways and the inferred grouping structure, our results shown in the left panel (with all treatments) of Figure 5 is consistent with the biological results in Rashid et al. (2021) mentioned above. The right panel (without combination treatments) of Figure 5 is our new discovery about the PDX data. We will look through further references to explore the biological interpretations for single treatments.

---

> > ### Comment · Reviewer_cwpD · 2022-08-08
> > **Reply**
> >
> > Thank you for your comments clarifying the PDX data and group lasso application. The additional group lasso results is a welcome addition to the supplement and also feel it does not need to be in the main manuscript.

---

> > > ### Author Response · Authors · 2022-08-08
> > > **Reply to the comments**
> > >
> > > We appreciate your response and acknowledgement of our clarifications for the paper. Thanks for your further suggestions about the group lasso step. As you suggested, to better clarify the group lasso step, we will add some results in the supplements.

---

### Official Review · Reviewer_NJYx · 2022-07-10

**Rating:** 5
**Confidence:** 3
**Soundness:** 3 good
**Presentation:** 3 good
**Contribution:** 3 good

**Summary:**

This paper proposed a method for learning the optimal individualized treatment rule (ITR). It focuses on a situation of having extensive treatment options but limited observations for only a small number of specific treatments. This work designs an algorithm to merge similar treatments and provide optimal ITR in a reduced treatment space. The model is trained based on a designed convex optimization problem with an adaptive proximal gradient algorithm. The empirical study reveals its ability to detect the structure of the treatment space and merge similar ones to enable the optimal ITR from many treatment options. A theoretical analysis is also provided.

**Questions:**

Referred to Strengths and Weaknesses

**Limitations:**

Will such a treatment algorithm introduce any bias to the patients?

**Strengths And Weaknesses:**

The algorithm focuses on a more practical situation of optimal ITR in precision medicine. The author models the problem as a convex minimization problem and solves it with an accelerated gradient method. This algorithm can provide a hierarchical structure of the relationship between treatments. By merging similar treatments, users can give optimal ITR on a reduced treatment space which is more feasible in practice.

I have the following concerns:
1. It looks like the final result automatically generates the hierarchical structure of the treatments. However, I didn't understand how such a hierarchical structure could happen by just solving the convex minimization problem. Your algorithm seems can only provide the proximity/dissimilarity matrix on treatments. Did you later apply any linkage-based hierarchical clustering algorithms on the proximity/dissimilarity matrix to generate the structure? Please specify.
2. Is there an analysis of your algorithm's time/space complexity?
3. Did you compare with other methods that can also provide a dendrogram?
4. In lines 170-171, should it be if \hat{\xi}_k = 0, then Z_k is classified as V?

---

> ### Author Response · Authors · 2022-08-01
> **Reply to the comments**
>
> Thanks for your nice summary of the paper and constructive comments. For your questions, please see the following clarifications.
>
> 1. For your comment about the hierarchical structure of the treatments, our algorithm does not use some two-step procedures. Instead, we automatically generate the dendrogram of the treatments by running our algorithm using different tuning parameters as a $\textbf{solution path}$. Our dendrogram is different from the standard understanding of the dendrogram generated from some specific hierarchical clustering algorithms. In contrast, in our Figures 2 and 5, the $y$ axis corresponds to the tuning parameter $\lambda$ rather than an explicit measure of “closeness” among treatments. Recall that the $\lambda$ showing up in the penalty term from Equations (2), (4) and (7) would encourage the treatments with similar treatment effects to merge into treatment groups. Thus, as $\lambda$ increases (corresponds to our dendrogram generating from bottom to top), the treatment structure will change from “no structure” (each treatment themselves is a treatment group, $\lambda = 0$), to the structure that all treatments are merged into one group ($\lambda \to \infty$). For each fixed $\lambda$, the treatment structure can be directly recovered by the group structure of the estimated parameter $\widehat{\mathbf{\beta}}$ in Equation (7). Therefore, our dendrogram can be better interpreted as the $\textbf{solution path}$ of the treatment clustering process (indicate which treatments are combined into one group when the turning parameter changes). More specifically, the whole solution path (dendrogram) can be automatically drawn using the solutions of Equation (7) as $\lambda$ increases. We will make this point more clear in the revision.
>
> 2. Thanks for pointing out the time/space complexity issue. Due to the usage of proximal gradient descent algorithm, the time and space complexities are both $\mathcal{O}(n^2)$, where $n$ is the training sample size. We will add this in the revision.
>
> 3. The comparison methods do not consider the group structure of the treatments. Hence, they cannot provide a dendrogram that demonstrates the solution path of the clustering process. Please refer to the details in our clarifications about your first concern above.
>
> 4. Yes. if the vector $\widehat{\mathbf{\xi}}_k = 0$, then $Z_k$ is classified as $V$.
>
> 5. Finally, we would like to discuss the issue of bias you mentioned in the limitations. (a) For the possible under representative issue in the training data, we can further improve our algorithm to protect fairness, i.e., recover the full representation for the target population, by incorporating some weights. (b) For the possible bias of the recommended treatment, recall that we merge the treatments into the same treatment groups because they have similar treatment effects and, hence they should be close to each other within the same group. As a result, the treatment effect bias should be small.

---

### Meta-Review · Area_Chair_wmCb · 2022-08-28

**Recommendation:** Accept
**Confidence:** Less certain

**Metareview:**

This paper proposes a method for learning the optimal individualized treatment rule (ITR). The proposed approach uses a fusion penalty term that encourages clustering between treatments. A dendrogram of the treatments is generated by running the proposed algorithm using different tuning parameters as a solution path. The effectiveness of the proposed approach is empirically validated on synthetic and real data. The paper is well written and technically sound. A thorough analysis/interpretation of the resulting model/results will further improve the paper.

**Award:**

No

---

### Decision · Program_Chairs · 2022-09-14

Accept